# SwitchLingua🗣️: The First Large-Scale Multilingual and Multi-Ethnic Code-Switching Dataset

**Peng Xie**[*]    **Xingyuan Liu**    **Yequan Bie**    **Tsz Wai Chan**
**Yangqiu Song**    **Yang Wang**    **Hao Chen**    **Kani Chen**[*]

The Hong Kong University of Science and Technology

## Abstract

Code-switching (CS) is the alternating use of two or more languages within a conversation or utterance, often influenced by social context and speaker identity. This linguistic phenomenon poses challenges for Automatic Speech Recognition (ASR) systems, which are typically designed for a single language and struggle to handle multilingual inputs. The growing global demand for multilingual applications, including Code-Switching ASR (CSASR), Code-Switching Text-to-Speech (CSTTS), and Cross-Lingual Information Retrieval (CLIR), highlights the inadequacy of existing monolingual datasets. Although some code-switching datasets exist, most are limited to bilingual mixing within homogeneous ethnic groups, leaving a critical need for a large-scale, diverse benchmark akin to ImageNet in computer vision. To bridge this gap, we introduce **LinguaMaster**, a multi-agent collaboration framework specifically designed for efficient and scalable multilingual data synthesis. Leveraging this framework, we curate **SwitchLingua**, the first large-scale multilingual and multi-ethnic code-switching dataset, including: (1) 420K CS textual samples across 12 languages, and (2) over 80 hours of audio recordings from 174 speakers representing 18 countries/regions and 63 racial/ethnic backgrounds, based on the textual data. This dataset captures rich linguistic and cultural diversity, offering a foundational resource for advancing multilingual and multicultural research. Furthermore, to address the issue that existing ASR evaluation metrics lack sensitivity to code-switching scenarios, we propose the **Semantic-Aware Error Rate (SAER)**, a novel evaluation metric that incorporates semantic information, providing a more accurate and context-aware assessment of system performance. Benchmark experiments on SwitchLingua with state-of-the-art ASR models reveal substantial performance gaps, underscoring the dataset's utility as a rigorous benchmark for CS capability evaluation. In addition, SwitchLingua aims to encourage further research to promote cultural inclusivity and linguistic diversity in speech technology, fostering equitable progress in the ASR field.

 **LinguaMaster (Code):** github.com/Shelton1013/SwitchLingua

🤗 **SwitchLingua (Data):** SwitchLingua_text & SwitchLingua_audio

## 1    Introduction

Code-switching (CS) refers to the phenomenon where two languages are spoken in contact within one utterance, as seen in Cantonese-English [1], Arabic-English [2], and Hindi-English [3]. This practice

---

* Corresponding authors: pxieaf@connect.ust.hk, makchen@ust.hk

39th Conference on Neural Information Processing Systems (NeurIPS 2025) Track on Datasets and Benchmarks.

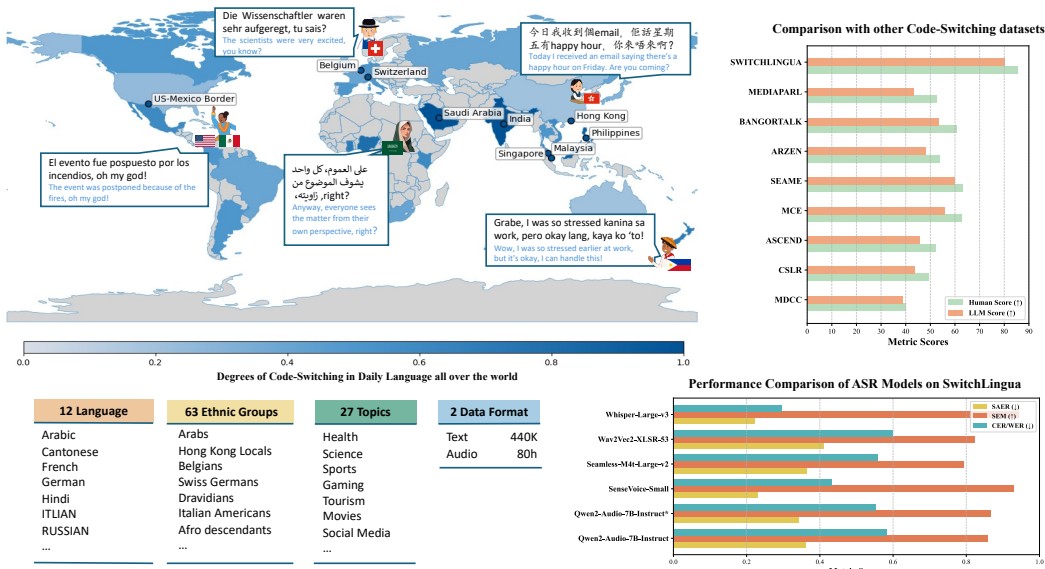

Figure 1: A summary of the SwitchLingua benchmark and the overall evaluation results. Our proposed SwitchLingua dataset includes code-switching textual and audio data spanning 12 languages and 63 ethnic groups, covering 27 topics. We also conduct comparison experiments with other code-switching datasets through both human and LLM evaluations to demonstrate the high quality and data richness of SwitchLingua. Benchmark experiments with state-of-the-art ASR models also indicate that SwitchLingua holds the potential to advance the research and applications of speech recognition and natural language processing. The upper-left part of this figure presents code-switching samples and the varying degrees of code-switching observed in daily life across different regions.

is prevalent in multilingual societies, as shown in Figure 1. As multilingualism and multiracialization become more common in today's globalized world [4], there has been increasing interest in Code-Switching Automatic Speech Recognition (CSASR).

Advancements in speech processing and natural language understanding heavily rely on the availability of large-scale audio datasets. However, most existing datasets are limited in terms of language diversity [5] and speaker representation [6]. Some datasets, sourced from online materials [7], exhibit low code-switching density due to the predominantly monolingual nature of the text. Others are artificially created by translating monolingual text into code-switching speech [8], but these translations frequently fail to capture the natural fluidity, spontaneity, and linguistic nuances of real-world code-switching, leading to unnatural and grammatically inconsistent speech. Such limitations hinder the development of robust automatic speech recognition models [9].

To address these challenges, synthetic data generation techniques [10] have emerged as a scalable and cost-effective alternative to manual data collection and annotation. By generating data using simulations, algorithms [8], or generative models [9], synthetic data provides an efficient and scalable solution to issues such as data scarcity, dataset imbalance, and privacy concerns. Unlike traditional methods [11, 12], which are often expensive, time-consuming, and fraught with legal or ethical constraints, synthetic approaches offer a flexible and cost-effective pathway to build large-scale, high-quality datasets.

While existing multilingual ASR models [13, 14] are trained on multiple languages, their final output is typically monolingual, limiting their ability to fully capture the linguistic diversity inherent in code-switching scenarios. This challenge is further exacerbated by the shortcomings of traditional evaluation metrics, such as Word Error Rate (WER) [15], Character Error Rate (CER) [16], and Match Error Rate (MER) [17]. These metrics are often overly rigid and insensitive to semantic equivalence, failing to account for differences in writing styles or transcription variations that do not alter the semantic meaning. For instance, variations in capitalization or transliteration can result in substantial

penalties under WER or CER, despite having minimal impact on the actual comprehension of the transcribed content [18]. In code-switching contexts, these issues become even more pronounced due to the inherent challenges in transcribing words from different languages. Differences such as the use of native scripts versus phonetic transliterations may lead to disproportionately high error rates, misrepresenting the true performance of ASR models. Therefore, there is an urgent need to develop specialized evaluation metrics tailored to code-switching, ensuring a more accurate assessment of ASR systems in multilingual and linguistically diverse contexts.

To overcome the above issues and to create a benchmark that holds the potential to advance the research and applications of speech recognition and natural language processing, in this paper, (i) We propose **a novel data synthesis framework: LinguaMaster**. It addresses the lack of high-quality code-switching data by aligning the synthesized data aligns with real-world linguistic patterns, providing a scalable solution for training robust ASR models. (ii) We create **the first large-scale code-switching dataset: SwitchLingua.** It serves as a critical resource for advancing ASR research in multilingual and code-switching scenarios. (iii) We introduce **a new evaluation metric: Semantic-Aware Error Rate (SAER)**. By emphasizing semantic understanding, SAER provides a more semantically sensitive and meaningful evaluation, accurately reflecting an ASR model's ability to handle complex linguistic phenomena and preserve the intended meaning of spoken content.

## 2 Related work

### 2.1 Code-Switching Data Synthesis

The demand for high-quality code-switching datasets remains critical across Code-Switching Automatic Speech Recognition (CSASR), Code-Switching Text-to-Speech (CSTTS), and Cross-Lingual Information Retrieval (CLIR) research and applications [14]. The success of modern AI and Computer Vision (CV), driven by large-scale benchmarks like ImageNet [19], underscores the importance of such resources in advancing research. Using machine translation (MT) [8] to translate specific segments within a monolingual text into another language is a widely adopted method for generating code-switching data. While efficient and scalable, MT-generated data often exhibits unnatural transitions, disrupted word order, and grammatical inconsistencies, limiting its linguistic authenticity. Alternatively, approaches based on large language models (LLMs) [9] have shown promise in capturing nuanced sociolinguistic patterns, yet their outputs frequently suffer from high redundancy and constrained diversity, reducing their utility for robust model training. To overcome these limitations, our work introduces a linguistically guided synthesis framework that integrates external tools and CS-specific constraints. This approach enhances grammatical accuracy, diversity, and naturalness while minimizing redundancy, enabling the generation of higher-quality CS data for downstream applications and paving the way for advancements in code-switching research.

### 2.2 LLM-based Multi-agent System

LLM-based Multi-agent System [20, 21] aims to enhance the reusability of implemented agents, facilitating more efficient and scalable development. For example, frameworks like CAMEL [22] employ cooperative autonomous agents to complete complex tasks with minimal human intervention. Further simplifying development, AutoGen [23] streamlines and consolidates multi-agent workflows through conversations, significantly reducing the effort required to build LLM applications across diverse domains. Compared to relying on a single LLM to handle complex tasks [24], multi-agent systems exhibit superior efficiency and accuracy across various domains and applications, such as role-playing, decision-making, and problem-solving. This cooperative framework represents a transformative advancement in the development of LLM-based applications, enabling more robust, adaptable, and efficient solutions across a wide range of use cases. In this paper, our framework harnesses the power of multi-agent collaboration, achieving higher-quality and more efficient data synthesis through optimized agent coordination.

### 2.3 Evaluation Metrics for Code-switching

In the context of CSASR, evaluation still predominantly relies on standard ASR metrics such as Word Error Rate (WER) [15] and Character Error Rate (CER) [16]. While these metrics provide a reasonable approximation of performance quality, they fail to account for semantic equivalence or al-

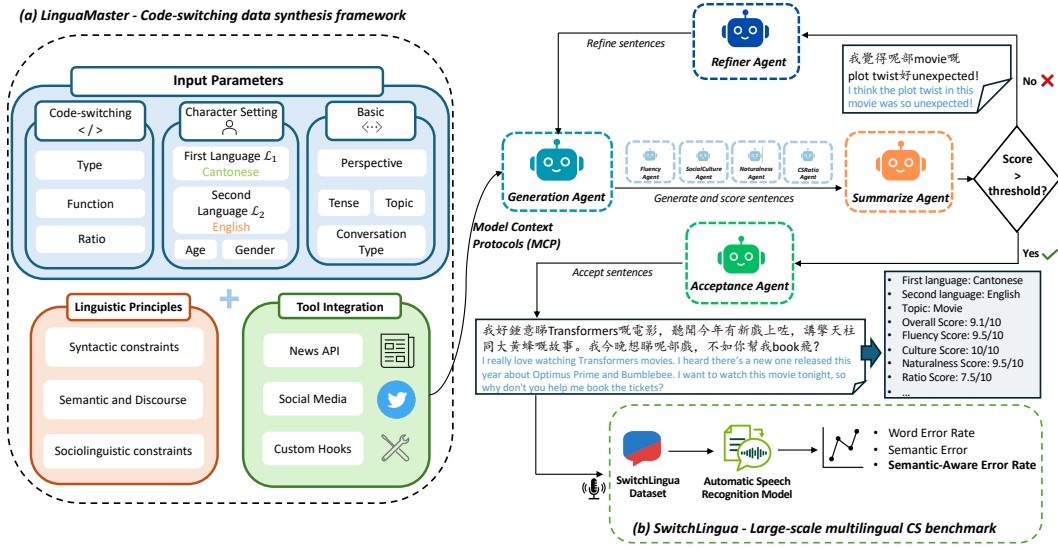

Figure 2: The pipeline of the **LinguaMaster** Framework. The *GenerationAgent* produces an initial code-switching sentence based on input parameters with the guidance of linguistic principles and the help of tools. Four linguistic evaluators independently score the sentence from different linguistic dimensions. The *SummarizeAgent* aggregates the evaluations and determines whether to let the *AcceptanceAgent* approve it or let the *RefinerAgent* re-enter the loop again and iteratively refine the sentence. All accepted data constitute the final **SwitchLingua**, serving as a large-scale multilingual and multi-ethnic code-switching benchmark dataset.

ternative valid transcriptions. This leads to a critical limitation in multilingual settings where multiple valid ways of transcribing or translating a phrase may exist while preserving the intended meaning. To alleviate the above issues, Mixed Error Rate (MER) [17] is introduced to accommodate differences in lexical units between Mandarin and English, while PolyWER [18] handles Arabic-English scenarios by accepting alternative forms of transcriptions. The multi-reference WER (mrWER) [25] framework further extends this concept for dialectal Arabic evaluation through multiple transcription references. However, these approaches share significant practical constraints: they require labor-intensive preparation of multiple reference transcriptions while still failing to comprehensively cover all potential variations. In this paper, we overcome these limitations by proposing an efficient and effective approach that directly integrates semantic similarity metrics into traditional evaluation metrics for code-switching scenarios, providing a more accurate and flexible evaluation framework.

## 3 Method

### 3.1 Preliminaries: Linguistic Principles (LP)

Code-switching as a linguistic phenomenon has attracted substantial scholarly attention [26, 27, 28]. Since code-switching may occur at almost any level of discourse (morphological, lexical, phrasal, or sentential) and is influenced by factors such as language competence, pragmatic intent, and demographic background, it has been examined from virtually every sub-discipline of linguistics. These studies reveal *how*, *when*, and *where* code-switching arises, thereby informing the framework of our generation method. Among them, research on syntax, semantics, pragmatics, and sociolinguistics is most relevant for automatic code-switching text generation.

### 3.1.1 Syntactic constraints

According to Poplack [28], code-switching can be classified by its *switch boundary*:

• **Inter-sentential** — The speaker completes a clause in the first language ($L_1$) and begins the next clause in the second language ($L_2$), requiring only clause-level grammatical compatibility.

| Language | Content | | | | | | |
|---|---|---|---|---|---|---|---|
| **English** ($L1$) | I | told him | that | so that | he | would bring it | fast. |
| **Spanish** ($L2$) | (Yo) | le dije | eso | pa' que | (él) | la trajera | ligero. |
| **English-Spanish** | I | told him | that | **pa' que** | | **la trajera** | **ligero**. |

Table 1: An illustrative CS example with syntactic constraint. "(Yo)" indicates that while "I" corresponds to "Yo" in Spanish, the pure Spanish sentence omits "Yo" as it is implied by the conjugated verb "le dije". Similarly, "(él)" serves the same function for the third-person pronoun.

| DATASET | HUMAN SCORE (↑) / LLM SCORE (↑) | | | | | | | Overall |
|---|---|---|---|---|---|---|---|---|
| | Linguistic Richness | Language and Rac. | Realism | Switching Nat. | Contextual Coh. | Grammatical Accuracy | Audio Quality | |
| MDCC[12] | 5.0/8.0 | 7.1/6.0 | 7.4/8.7 | 5.4/6.0 | 4.4/6.0 | 5.6/4.0 | 5.2/- | 40.1/38.7 |
| CSLR [6] | 6.3/8.7 | 5.2/6.0 | 8.1/9.0 | 6.4/5.0 | 7.0/7.0 | 7.5/8.0 | 8.9/- | 49.4/43.7 |
| ASCEND [11] | 6.7/8.3 | 7.2/6.0 | 8.3/10.0 | 6.7/7.0 | 7.4/8.3 | 6.9/6.0 | 9.0/- | 52.2/45.6 |
| MCE [29] | 11.4/11.3 | 8.4/10.3 | 12.3/11.7 | 6.8/7.3 | 7.4/7.7 | 7.1/7.7 | 9.3/- | 62.7/56.0 |
| SEAME [30] | 12.1/13.0 | 10.7/13.7 | 9.0/11.0 | 6.0/7.3 | 8.5/7.0 | 7.6/8.0 | 9.3/- | 63.2/60.0 |
| ARZEN [31] | 6.5/9.0 | 8.0/10.0 | 10.1/9.0 | 6.0/7.3 | 7.2/6.0 | 7.0/7.0 | 9.1/- | 53.9/48.3 |
| BANGORTALK[7] | 7.8/9.0 | 12.4/14.3 | 11.5/11.0 | 6.4/7.0 | 7.3/6.0 | 7.2/6.0 | 8.1/- | 60.7/53.3 |
| MEDIAPARL[32] | 8.1/10.0 | 9.7/6.0 | 5.2/7.0 | 6.3/7.3 | 7.7/7.0 | 6.4/6.0 | 9.2/- | 52.6/43.3 |
| **SWITCHLINGUA (ours)** | **16.2/17.7** (↑25.9%) | **18.1/19.7** (↑31.0%) | **16.4/17.0** (↑28.1%) | **7.2/7.6** (↑4.8%) | **9.2/9.7** (↑12.3%) | **8.8/8.3** (↑8.6%) | **9.5/-** (↑2.1%) | **85.4/80.0** (↑25.5%) |

Table 2: Comparison of human and LLM-based evaluation scores across various cs datasets. The metrics include Linguistic Richness (20ps), Language and Racial Diversit (20ps), Realism (20ps), Switching Naturalness (10ps), Contextual Coherence (10ps), Grammatical Accuracy (10ps) and Audio Quality (10ps). The Overall column reports the total score. Note that since LLMs are not well-suited for evaluating audio quality, the LLM-based overall score is calculated out of 90 points. The best results are in **bold**. The red text highlights the average percentage (Human and LLM Score) by which the SwitchLingua dataset outperforms the second highest-scoring dataset in each metric.

- **Extra-sentential** — Tags, fillers, or exclamations in $L_2$ are appended to an otherwise monolingual $L_1$ utterance, still governed by clause-level grammar.

- **Intra-sentential** — The switch occurs *within* a sentence, spanning a morpheme, phrase, or embedded clause, demanding stronger syntactic compatibility for acceptability.

Table 1 shows an illustrative alignment example with syntatic constraints. For the intra-sentential case, we adopt Poplack's general constraints [28]:

(a) **Free–Morpheme Constraint.** *Switches may occur after any constituent that is not a bound morpheme.* A bound morpheme cannot be stranded.

(b) **Equivalence Constraint.** *A switch tends to occur only at points where the surface word-order of $L_1$ and $L_2$ coincide*, so that no monolingual rule is violated. In practice we align an $L_1$ sentence with its $L_2$ translation and mark all overlapping positions as permissible switch points.

### 3.1.2   Semantic and conversation aspects

Code-switching is often influenced by semantic content and discourse context [37, 38, 27], as bilinguals switch languages to better express specific concepts or nuances. Certain terms or ideas may be more precise or culturally appropriate in one language, leading speakers to switch for clarity or emphasis. For example, a bilingual might use a technical term in the formal language of schooling or switch to their native language for everyday matters. Idioms, sayings, or culturally specific terms are also common triggers, as their meaning would be diluted in translation. Myers-Scotton highlights that code-switching often occurs because the alternate language better conveys the speaker's semantic or pragmatic intentions [27]. This practice enhances expressive precision, clarifies meaning, and allows bilinguals to fully utilize the richness of their language repertoire.

| LANGUAGE | EVALUATION METRICS | MODEL | | | | | |
|---|---|---|---|---|---|---|---|
| | | Qwen2-Audio-7B-Instruct [33] | Qwen2-Audio-7B-Instruct* [33] | SenseVoice-Small [34] | Seamless-M4T-Large-v2 [35] | Wav2Vec2-XLSR-53 [36] | Whisper-Large-v3 [14] |
| ARABIC ENGLISH | CER ($\downarrow$) | 0.7880 | 0.7749 | - | 0.4633 | 0.8321 | **0.3406** |
| | SEM ($\uparrow$) | 0.7045 | 0.7076 | - | **0.9188** | 0.7429 | 0.8846 |
| | SAER ($\downarrow$) | 0.5417 | 0.5337 | - | 0.2723 | 0.5446 | **0.2280** |
| CANTONESE ENGLISH | CER ($\downarrow$) | 0.4599 | 0.4137 | **0.2440** | 0.3975 | 0.6330 | 0.3838 |
| | SEM ($\uparrow$) | 0.9230 | 0.9385 | **0.9668** | 0.9346 | 0.8246 | 0.9537 |
| | SAER ($\downarrow$) | 0.2685 | 0.2376 | **0.1386** | 0.2314 | 0.4042 | 0.2187 |
| FRENCH ENGLISH | WER ($\downarrow$) | 0.4661 | 0.4442 | 0.4224 | 0.7088 | 0.4918 | **0.2483** |
| | SEM ($\uparrow$) | 0.8939 | 0.8987 | 0.9287 | 0.6688 | 0.8398 | **0.9316** |
| | SAER ($\downarrow$) | 0.2861 | 0.2728 | 0.2468 | 0.5200 | 0.3260 | **0.1584** |
| GERMAN ENGLISH | WER ($\downarrow$) | 0.5699 | 0.5084 | 0.3825 | 0.7207 | 0.4855 | **0.2568** |
| | SEM ($\uparrow$) | 0.8712 | 0.8864 | 0.9365 | 0.6854 | 0.8525 | **0.9452** |
| | SAER ($\downarrow$) | 0.3493 | 0.3110 | 0.2230 | 0.5177 | 0.3165 | **0.1558** |
| HINDI ENGLISH | WER ($\downarrow$) | 0.6953 | 0.6570 | 0.5546 | 0.4744 | 0.6029 | **0.2697** |
| | SEM ($\uparrow$) | 0.7627 | 0.7696 | 0.8071 | 0.8188 | 0.7597 | **0.9458** |
| | SAER ($\downarrow$) | 0.4663 | 0.4437 | 0.3738 | 0.3278 | 0.4216 | **0.1991** |
| ITALIAN ENGLISH | WER ($\downarrow$) | 0.5949 | 0.5701 | 0.2881 | 0.6652 | 0.3961 | **0.2498** |
| | SEM ($\uparrow$) | 0.8620 | 0.8677 | **0.9650** | 0.6932 | 0.8897 | 0.9305 |
| | SAER ($\downarrow$) | 0.3664 | 0.3512 | 0.1615 | 0.4860 | 0.2532 | **0.1596** |
| JAPANESE ENGLISH | CER ($\downarrow$) | 0.5423 | 0.4762 | 0.4355 | 0.5277 | 0.7336 | **0.3258** |
| | SEM ($\uparrow$) | 0.8833 | 0.8941 | 0.9225 | 0.9097 | 0.7469 | **0.9344** |
| | SAER ($\downarrow$) | 0.3295 | 0.2910 | 0.2565 | 0.3090 | 0.4934 | **0.1957** |
| KOREAN ENGLISH | CER ($\downarrow$) | 0.7109 | 0.6900 | 0.4717 | 0.4013 | 1.0000 | **0.1143** |
| | SEM ($\uparrow$) | 0.7019 | 0.7058 | 0.8870 | 0.8622 | 0.4660 | **0.9458** |
| | SAER ($\downarrow$) | 0.5045 | 0.4921 | 0.2924 | 0.2696 | 0.7670 | **0.8561** |
| MANDARIN ENGLISH | CER ($\downarrow$) | 0.5068 | 0.4852 | 0.4008 | **0.0700** | 0.6433 | 0.1703 |
| | SEM ($\uparrow$) | 0.9319 | 0.9354 | 0.9697 | **0.9892** | 0.8821 | 0.9718 |
| | SAER ($\downarrow$) | 0.2875 | 0.2749 | 0.2156 | **0.0404** | 0.3806 | 0.0992 |
| RUSSIAN ENGLISH | WER ($\downarrow$) | 0.6278 | 0.6091 | 0.4364 | 0.7057 | 0.6358 | **0.1354** |
| | SEM ($\uparrow$) | 0.8218 | 0.8263 | 0.9179 | 0.6427 | 0.7929 | **0.9653** |
| | SAER ($\downarrow$) | 0.4030 | 0.3914 | 0.2593 | 0.5315 | 0.4215 | **0.0950** |
| SPANISH ENGLISH | WER ($\downarrow$) | 0.3223 | 0.2978 | 0.2437 | 0.6978 | 0.2859 | **0.1293** |
| | SEM ($\uparrow$) | 0.9417 | 0.9475 | 0.9640 | 0.6673 | 0.9134 | **0.9874** |
| | SAER ($\downarrow$) | 0.1903 | 0.1752 | 0.1399 | 0.5152 | 0.1863 | **0.0707** |

Table 3: Performance comparison of various ASR models across multiple languages. The evaluation metrics include CER/WER, SEM, and our proposed SAER. Qwen2-Audio-7B-Instruct* represents the cleaned outputs, as this model's raw outputs may include additional information that impacts scoring. The best results are in **bold**.

CS not only adheres to syntactic rules but also enriches meaning through the interplay of languages. Switching can convey subtext, signal shifts in topic or speaker stance or act as a discourse strategy, as Gumperz [38] described, functioning as a metaphorical "cue". For example, one language might frame serious discussion, while another conveys humor or intimacy. A single word from $L_2$ within an $L_1$ sentence can evoke the cultural and conceptual associations of $L_2$, embedding its context into the conversation. Code-switching thus allows bilinguals to weave the semantic richness of both languages into their discourse, enhancing meaning and cultural resonance. See the Appendix D for more detailed linguistic analysis.

### 3.1.3 Sociolinguistic constraints

In addition to structural and semantic factors, CS is deeply rooted in sociolinguistic context [39, 40, 41]. Various factors including gender, age, and ethnicity influence the use of CS, while the location, situation, register, and partner affect each conversation. A bilingual or multilingual person's language

choice – the decision to speak a certain language in a certain situation – is often politically, socially, or personally motivated, which can be applied to the choice of using CS. CS can be a collective behavior. Societal bilingualism or multilingualism refers to the phenomenon where the use of two or more languages is a norm for a group of people. In India, for example, many people speak their regional language as well as Hindi, the most widely spoken of the country's Indigenous official languages. Most educated speakers additionally speak English, which is also an official language of India. Heller shows that speakers use CS to negotiate power and identity [41]. Our generation framework therefore incorporates demographic and situational metadata to emulate authentic bilingual interaction.

## 3.2 The LinguaMaster Framework

**Multi-agent collaboration.** An overview of our proposed code-switching data synthesis framework LinguaMaster is illustrated in Figure 2. Multiple LLM-based agents collaborate to improve the accuracy and efficiency of various tasks [20, 21]. However, these LLMs often lack an inherent understanding of the linguistic principles governing code-switching. By incorporating the code-switching linguistic rules discussed in Section 3.1 into the LLMs, we can significantly enhance the coherence and authenticity of the generated data in terms of language transitions. Additionally, integrating external tools with LLMs can further enrich the diversity of the synthesized data while reducing redundancy, creating a more robust and representative dataset for downstream applications. Specifically, LinguaMaster follows a **generate–evaluate–refine** ideology (refer to the Appendix B.1 for detailed descriptions of each agent):

(1) **GenerationAgent** produces an initial code-switched candidate.

(2) Four linguistic evaluator agents assess the candidate, including **FluencyAgent**, **NaturalnessAgent**, **CSRatioAgent**, and **SocialCultureAgent**.

(3) **SummarizeAgent** aggregates the evaluation scores to determine: (i) **AcceptanceAgent** finalizes high-quality outputs, or (ii) **RefinerAgent** iteratively improves suboptimal candidates.

**Linguistically constrained generation.** *GenerationAgent* is not a free-form LLM sampler. Before emitting a candidate sentence, it follows the four-step Structure & Switch routine:

(1) Parse the $L_1$ sentence into a dependency tree (i.e., a hierarchical representation of the syntactic structure of a sentence[1]) to identify different syntactic components.

(2) Translate it to $L_2$ and obtain token alignment, as illustrated in Table 1.

(3) Locate all switch points that respect (a) the Free-Morpheme and (b) the Equivalence constraints.

(4) sample one permissible span and splice the $L_2$ fragment back into the $L_1$ skeleton.

The generator's search space is filtered by these syntactic rules, and every output is guaranteed clause-internal well-formedness in both languages.

**Tool Integration (TI) via Model Context Protocols (MCP).** Before sentence synthesis begins, *GenerationAgent* optionally invokes a suite of MCP to pull fresh, domain–relevant snippets. The MCP layer serves as a thin shim that connects LinguaMaster with various external information sources. Each MCP tool processes a user-level topic query and defines a lightweight parser to convert the retrieved snippet into textual context blocks, which are then appended to the LLM prompt. The MCP layer is *tool-agnostic*, with each external resource encapsulated behind a trivial `ToolSpec` interface: `ToolSpec` = $\langle$Name, Auth, Query$(\cdot)$, Parse$(\cdot)\rangle$, enabling declarative registration and hot-swappable integration. Note that when the MCP layer provides topical snippets, the same constraint pipeline is applied: the snippet is paraphrased into $L_1$, aligned, and then mixed, ensuring that information enrichment never violates the linguistic principles defined in Section 3.1. In this way, LinguaMaster seamlessly combines the rich external context of modern tool-augmented LLMs with the classic grammatical rigor of CS research. More implementation details and algorithm workflow can be found in the Appendix B.1.

## 3.3 Semantic-Aware Error Rate Evaluation Metric

To provide a more semantically sensitive and meaningful evaluation, we propose Semantic-Aware Error Rate (SAER). This metric incorporates the multilingual semantic similarity model LaBSE [42]

---

[1]https://en.wikipedia.org/wiki/Dependency_grammar

into traditional ASR evaluation metrics [15, 16], focusing not only on word/character-level errors but also semantic consistency. Specifically, we first define the semantic error $\varepsilon_{\text{sem}}$, which measures the dissimilarity between the predicted sentence $\hat{y}$ and the reference sentence $y$ in semantic space, and it is computed using cosine distance:

$$\varepsilon_{\text{sem}} = 1 - \frac{f(\hat{y})^\top f(y)}{\|f(\hat{y})\| \cdot \|f(y)\|}, \tag{1}$$

where $f(\cdot)$ represents the semantic embedding function that maps sentences into a vector space. Then let $\mathcal{F}$ denote the language-specific form error, which accounts for character-level (CER) and word-level (WER) discrepancies between $\hat{y}$ and $y$:

$$\mathcal{F} = \begin{bmatrix} \text{WER}(\hat{y}, y) \\ \text{CER}(\hat{y}, y) \end{bmatrix} \quad \boldsymbol{\delta}(\lambda(y)) = \begin{bmatrix} \mathbf{1}_{\mathcal{L}_1}(\lambda(y)) \\ \mathbf{1}_{\mathcal{L}_2}(\lambda(y)) \end{bmatrix}, \tag{2}$$

where $\lambda(y)$ denotes the matrix language of reference $y$, and the indicator vector $\boldsymbol{\delta}(\lambda(y))$ ensures only one error metric is activated per input ($\mathbf{1}_{\mathcal{L}}(\cdot)$ is the indicator function). $\mathcal{L}_1$ corresponds to logographic language space (e.g., Simplified Chinese, Traditional Chinese, Japanese), while $\mathcal{L}_2$ includes alphabetic language space (e.g., English, German, French).

The final SAER metric combines the semantic error and language-specific form error into a weighted equation:

$$\text{SAER}_\alpha(\hat{y}, y) = (1 - \alpha)\, \varepsilon_{\text{sem}} + \alpha \cdot \langle \boldsymbol{\delta}(\lambda(y)),\ \mathcal{F} \rangle, \tag{3}$$

where $\alpha$ is a tunable parameter that balances the contribution of semantic and form errors. The inner-product term $\langle \boldsymbol{\delta}(\lambda(y)),\ \mathcal{F} \rangle$ ensures that language-specific differences are appropriately weighted in the evaluation.

## 4 Experiments

### 4.1 Experimental Setup

**Dataset.** The SwitchLingua encompasses 12 languages, 63 different ethnic groups, 9 major topics, and 27 subcategories, including both text and audio data modalities. The dataset is organized into two forms: single-turn dialogues and multi-turn dialogues. More details about the dataset are in the Appendix A.3.

**Implementation details.** To evaluate the quality of SwitchLingua, we employ both human evaluation and LLM evaluation. For human evaluation, the evaluators are native speakers of the relevant languages to ensure linguistic accuracy and cultural appropriateness of the audio data. For LLM evaluation, we leverage GPT-4o [43] as the evaluation model [44] for the textual data, capitalizing on its advanced language understanding capabilities. Additionally, we position SwitchLingua as a benchmark for assessing the performance of existing ASR models [34, 35, 33, 36, 45, 14] in CS scenarios, which demonstrates that CS remains a significant challenge for current ASR systems, highlighting the need for further research and development in this field. LLM agents are based on GPT-4o [43] with different input prompts. Experiments are conducted on RTX A6000 GPUs. More details are in the Appendix B.1.

### 4.2 Experimental Results

**Validating SwitchLingua with human and LLM-based judgments.** As shown in Table 2, we conduct a comprehensive evaluation to assess how closely our synthetic dataset aligns with naturally occurring human language data. The evaluation benchmark datasets includes MDCC [12], CSLR [6], ASCEND [11], MCE [29], SEAME [30], ArzEn [31], BangorTalk [7], and MediaParl [32]. To ensure accurate and realistic evaluation, we engage native speakers from 40 distinct linguistic and ethnic backgrounds to assess the audio data quality across seven key dimensions (see appendix C.1 for scoring details). Additionally, GPT-4o [43] is employed to evaluate the quality of the corresponding code-switching textual data to make the evaluation more robust. It can be observed that SwitchLingua respectively outperforms the second-best dataset by 4.1/4.7, 5.7/5.4, 4.1/5.3, 0.4/0.3, 0.7/1.5, 1.2/0.3, and 0.2/- in the seven evaluation metrics, demonstrating its comprehensive superiority. Notably,

| Method | Human Score (↑) / LLM Score (↑) | | | | | |
|---|---|---|---|---|---|---|
| | Linguistic Richness | Realism | Switching Naturalness | Contextual Coherence | Grammatical Accuracy | Overall |
| Baseline | 6.3/8.0 | 9.7/11.0 | 5.0/5.3 | 5.3/6.0 | 8.3/9.3 | 34.6/39.6 |
| + LP | 7.7/9.3 | 11.3/12.0 | 6.7/8.0 | 6.7/7.0 | 8.7/9.7 | 41.1/46.0 |
| + LP + MAC | 12.3/12.0 | 15.3/14.7 | 7.7/8.3 | 7.0/7.3 | 9.3/9.7 | 51.6/52.0 |
| **+ LP + MAC + TI** | **17.0/17.7** (↑**58.2%**) | **16.7/17.7** (↑**39.9%**) | **9.3/9.7** (↑**46.6%**) | **9.3/10.0** (↑**41.5%**) | **9.7/10.0** (↑**8.7%**) | **62.0/65.1** (↑**41.7%**) |

Table 4: Ablation study of our LinguaMaster framework. MAC, LP and TI represent the Multi-Agent Collaboration, Linguistic Principle, and Tool Integration, respectively. The full score is 70 points and the improvements compared to the baseline are highlighted.

*Linguistic Richness*, *Language & Racial Diversity*, and *Realism* are three critical factors for addressing diverse and complex multilingual environments. SwitchLingua gains relative improvement of 25.9%, 31.0%, and 28.1% over the second-best results under these three metrics, respectively. These results highlight SwitchLingua's superior diversity and authenticity in multilingual data generation.

**SwitchLingua works as a multilingual CSASR benchmark.** The evaluation results for the ASR models are shown in Table 3. The evaluated models include Qwen2-Audio-7B-Instruct [33], SenseVoice-Small [34], Seamless-M4T-Large v2 [35], Wav2Vec2-XLSR-53 [36], and Whisper-Large-v3 [14]. For models that require custom textual instruction (e.g., Qwen2-Audio-7B-Instruct), we use *"Only output what this person said"*. Nevertheless, some extraneous outputs still occur, such as *"The original content of this audio is:"*, *"The speaker said:"*, or *"The audio states:"*. We clean these extraneous outputs, and the average WER and SAER saw significant reductions of 6.75% and 5.34%, respectively. This demonstrates that WER or CER places a high demand on textual consistency, as even a small amount of irrelevant information can cause a significant increase in error rates. From the evaluation results, we find that Whisper-Large-v3 is the most effective ASR model, achieving the best SAER results in 8 languages. Some models such as Qwen2-Audio-7B-Instruct, score relatively low on the CSASR task due to language switching errors. Specifically, Qwen2-Audio-7B-Instruct exhibits a tendency to overproduce Chinese text, even when the target language is not Chinese. This issue likely stems from interference in language detection during code-switching scenarios.

**Ablation study.** We conduct ablation studies to demonstrate the effectiveness of each proposed module in LinguaMaster, as shown in Table 4, where GPT-4o is adopted as the baseline. Specifically, all the proposed components improve the synthetic data quality. More ablation results are in the Appendix B.2.

## 5 Conclusion

In this work, we propose *LinguaMaster*, a novel data synthesis framework that generates high-quality code-switching data aligned with real-world linguistic patterns. Leveraging this framework, we curate *SwitchLingua*, the first large-scale multilingual and multiracial dataset in the code-switching domain. Additionally, to address the limitations of traditional ASR metrics, we introduce **Semantic-Aware Error Rate**, which is a new evaluation metric that prioritizes semantic understanding and enables more accurate and semantic-sensitive evaluation. We envision LinguaMaster as a reference for future data synthesis methodologies and anticipate that SwitchLingua will work as a benchmark that accelerates advancements in the research and development of code-switching automatic speech recognition and natural language processing.

**Limitations**. Although our dataset has the potential to advance the development of the code-switching automatic speech recognition and natural language processing field, it also poses a risk of being misused for voice synthesis in fraudulent schemes. [2]

---

[2]We require users to sign licensing agreements or terms of use that explicitly restrict misuse, such as using the dataset for fraudulent purposes (e.g., voice synthesis for impersonation or scams), and specify acceptable use cases (e.g., research on automatic speech recognition or natural language processing) and prohibit unethical applications.

## Acknowledgments and Disclosure of Funding

This work was supported by grant number MOST24SC01 and RGC grant 16310222. We also thank the Crypto-Fintech Lab at the Hong Kong University of Science and Technology for their support, as well as the students who helped us collect the SwitchLingua dataset.

# 6 Licensing Agreement for the Use of SwitchLingua Dataset

This Licensing Agreement is entered into by and between the authors of the SwitchLingua dataset and the user or organization accessing the datase. By accessing, downloading, or using the dataset, the Licensee agrees to comply with the terms and conditions set forth below.

## 6.1 Permitted Use

The SwitchLingua dataset is provided for research and non-commercial purposes only. Permissible uses include, but are not limited to:

Developing, evaluating, or improving Automatic Speech Recognition (ASR), Text-to-Speech (TTS), and Cross-Language Information Retrieval (CLIR) systems. Academic research and publication, provided proper credit is given to the Licensor.

## 6.2 Prohibited Use

The Licensee agrees NOT to use the SwitchLingua dataset for any of the following purposes:

Fraudulent or malicious activities, including but not limited to voice synthesis for impersonation, scams, or identity theft. Development of systems intended to mislead or deceive individuals or organizations. Commercial applications without prior written permission from the Licensor. Redistribution, sublicensing, or resale of the dataset, in whole or in part.

## 6.3 Safety and Security Measures

To mitigate risks of misuse, the Licensee agrees to:

Use the dataset only within secure, controlled environments. Implement safeguards to prevent unauthorized access or sharing of the dataset. Disclose to the Licensor the intended use case and provide assurances of compliance with this Agreement.

## 6.4 Enforcement and Compliance

The Licensee agrees to:

Sign this Agreement as binding acknowledgment of compliance with its terms. Provide their institutional or organizational affiliation and contact details for accountability. Allow periodic audits by the Licensor to ensure compliance with the terms of this Agreement. In the event of a breach of this Agreement, the Licensor reserves the right to:

Revoke access to the dataset immediately. Pursue legal action, if necessary. Report misuse to relevant authorities.

## 6.5 Attribution

The Licensee must provide proper attribution to the Licensor in any research papers, publications, or derivative works that make use of the SwitchLingua dataset.

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

# A Code-switching Corpus and SwitchLingua

## A.1 Existing Open-source Corpus

Open-source code-switching corpora are highly valuable, yet many datasets remain either closed-source or no longer accessible. We focus on analyzing code-switching datasets that remained publicly available after March 2025. As summarized in Table A.1, the available datasets include ARZEN [31], ASCEND [11], BANGORTALK (includes SIARD, PATAGONIA, MIAMI) [7], CSLR [6], MCE [29], MEDIAPARL [32], and SEAME [30].

## A.2 Limitations of the Existing Corpus

To analyze the limitations of existing CS corpus, we use ASCEND [11], which is a Chinese-English CS dataset, to perform a case study. Table A.2 highlights the primary drawbacks of ASCEND. While its design of treating each row as an isolated speech slice offers convenience, it imposes significant research constraints, limiting applicability in more complex and context-dependent scenarios.

**Practical Implications:** Without span-level annotations or functional labels, models trained on ASCEND treat "mixed" as a monolithic class, preventing evaluation at exact switch points. Sociolinguistic hypotheses, e.g., whether the Matrix Language Frame holds, are likewise untestable.

## A.3 The SwitchLingua Dataset

SwitchLingua is a comprehensive multilingual and multicultural code-switching dataset designed to advance research in automatic speech recognition, natural language processing, and conversational AI. The textual data for SwitchLingua was first generated using the proposed LinguaMaster framework, and the audio data was recorded by 174 bilingual speakers from diverse linguistic and cultural backgrounds to ensure high quality. The final dataset comprises 420K textual samples and 80+ hours of recordings, making it the largest open-source code-switching dataset in terms of linguistic diversity and semantic richness.

As illustrated in Figure A.1, SwitchLingua incorporates 12 languages, including Arabic, Cantonese, French, etc., providing extensive coverage for cross-lingual and multilingual studies. Additionally, the dataset represents 63 ethnic groups, such as Kurds, Hong Kong locals, and Belgians, ensuring a broad spectrum of cultural and regional perspectives. The dataset spans 9 major topics, such as health, technology, and science, which are further divided into 27 subcategories, including cooking, social media, and artificial intelligence. This thematic structure supports fine-grained exploration of domain-specific conversational contexts.

To satisfy diverse research needs, SwitchLingua supports both single-turn and multi-turn dialogues, enabling the analysis of simple question-and-answer interactions as well as more complex, contextually rich discussions.

## A.4 SwitchLingua Data Sample

Figure A.2, A.3 and A.4 illustrate the structured analysis of code-switching text generation and evaluation results. Take Figure A.2 as an example, the text generation focuses on a first-person narrative on the topic of technology, set in the past tense, with a language mix of 50% Arabic and 50% English. The speaker is a female individual above 66 years old with a Master's education level, whose first language is Arabic and second language is English. The conversation type is multi-turn, and the code-switching type is intra-sentential with a metalinguistic function. The generated text demonstrates high fluency and naturalness, with scores of 9 in both categories, indicating that code-switching occurs seamlessly and authentically. Additionally, the language ratio analysis reveals that Arabic constitutes 63% of the text while English accounts for 37%, as Arabic serves as the matrix language. The social-cultural evaluation, with a score of 9, highlights that the use of English terms within an Arabic context is appropriate and reflects common linguistic practices. Overall, the generated text achieves a high overall score of 8.6, demonstrating the effectiveness of the code-switching model in producing contextually relevant multi-language narratives.

| Name | Data source | Size (hours) | Language | Speaker background |
|------|-------------|--------------|----------|---------------------|
| ARZEN [31] | Interviews | 12.0 | Egyptian Arabic English | Egypt |
| ASCEND [11] | Dialogues | 10.6 | Mandarin English | Hong Kong |
| SIARD [7] | Conversation | 40.8 | Welsh English | United Kingdom |
| PATAGONIA [7] | Conversation | 21.3 | Welsh Spanish | United Kingdom |
| MIAMI [7] | Conversation | 40.8 | Welsh Spanish | United Kingdom |
| CSLR [6] | Phone | 71.3 | Mandarin English | Mainland |
| MCE [29] | LLM | 34.8 | Cantonese English | HongKong GuangDong |
| MEDIAPARL [32] | Parliament | 40.0 | French German | Switzerland |
| SEAME [30] | Interview Conversation | 30.0 | Mandarin English | Singapore Malaysia |
| **SwitchLingua (audio)** | **LLM** | **80.2** | **Arabic Spanish French Mandarin Japanese Hindi German Italian Russian English Cantonese Korean** | **Kurds, Berbers, Arabs, Macau Locals and 59 other ethnic groups** |

Table A.1: Comparison of the existing code-switching datasets, with details of data sources, durations, covered languages, and speaker backgrounds. SwitchLingua-Audio covers the most languages and ethnic backgrounds.

# B  LinguaMaster Implementation Details

## B.1  Algorithm

LinguaMaster is implemented as a typed state graph, with vertices representing agent invocations and edges capturing data-flow or control-flow dependencies. Table B.1 outlines the detailed functions of the nine agents. The process begins with the *GenerationAgent*, which generates an initial code-switched candidate. Four linguistic evaluators then assess the candidate, followed by the *SummarizeAgent* aggregating the scores. Finally, the *AcceptanceAgent* either finalizes the sample, or the *RefinerAgent* iteratively refines it as needed.

Before the sentence synthesis starts, *GenerationAgent* optionally calls a suite of MCP to pull fresh, domain–relevant snippets. The MCP layer is a thin shim that bridges LinguaMaster and arbitrary external information sources. An MCP tool accepts a user-level `topic` query, returns a `JSON` snippet, and defines a light parser that converts the snippet into a textual "context block" appended to the LLM prompt. Internally each tool exposes four fields $\langle \text{Name}, \text{Auth}, \text{Query}(), \text{Parse}() \rangle$, making tool registration declarative (YAML) and hot-swappable.

**Built-in connectors.**

- **News API()** — grabs headline + lead paragraph from Reuters, BBC, Xinhua, etc.

| Issue | Implications |
|---|---|
| Utterance-level slices | The discourse context that triggers a switch is lost, making models hard to learn inter-sentential phenomena. |
| Only coarse "zh/en/mixed" labels | No token-level switch indice, making it impossible to evaluate or supervise fine-grained code-mixing. |
| No functional/constraint tags | Difficult to perform sociolinguistic analysis (directive, expressive . . . ) and constraint-aware generation. |
| Minimal speaker metadata | Cannot analyze proficiency, dialect, or demographic bias. |
| Skewed topic distribution | Over-representation of one topic (e.g., 26% samples are "technology") reduces domain diversity. |
| Many single-token turns | The high proportion of fillers and back-channels increases sparsity, leaving minimal syntactic signal. |
| Segmentation artifacts | Logical sentences are fragmented across rows (e.g., "we need smart" + "phone"). |
| No difficulty/ratio bins | Easy noun borrowings and complex intra-clause switches are conflated, with no structured progression. |
| Unknown language dominance | Chinese-dominant and balanced bilinguals are not distinguished, introducing bias into the models. |

Table A.2: Key limitations of existing code-switching corpus, taking ASCEND corpus as an example.

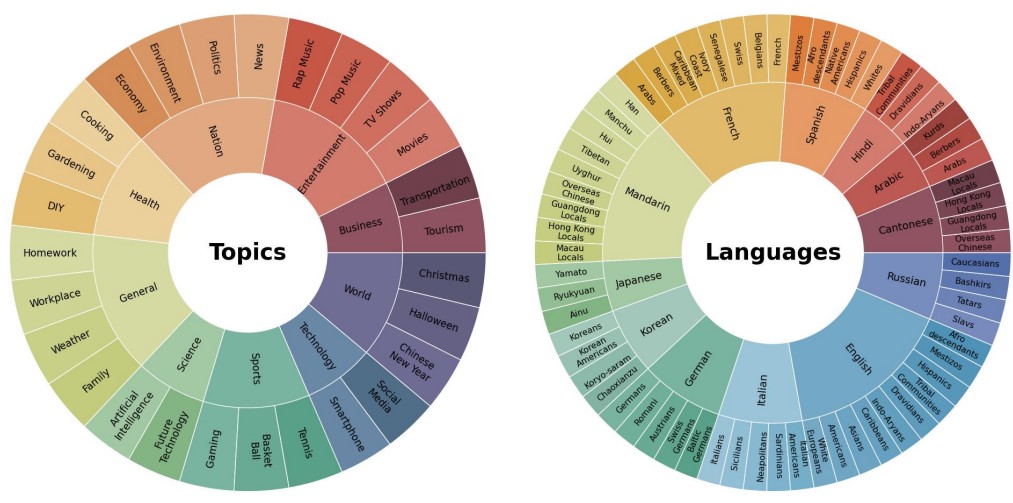

Figure A.1: Visualization of the covered languages and topics of the SwitchLingua dataset. The left chart illustrates the topical categories and their subtopics, while the right chart details the distribution of languages and ethnic groups, showcasing the dataset's diversity and richness.

- **Social Media()** — Discord channel history and the public X/Twitter search endpoint.
- **Custom Hooks()** — User scripts for niche forums, domain corpora, or proprietary KBs.

Because tools are decoupled from the state-graph, *any number* of MCP instances can be added at run-time. At inference we iterate over the current tool set $\mathcal{T} = \{T_1, \ldots, T_k\}$ and concatenate all retrieved snippets. If the aggregated context would exceed the model window, snippets are truncated or sub-sampled proportional to historical utility. This design yields a theoretically unbounded yet safely constrained information enhancement stage, allowing generated code-switched sentences to reflect up-to-date, real-world discourse.

The returned snippets are injected into the LLM prompt *as additional context*, yielding two benefits: (i) the lexical and topical distribution of the generated code-switched text closely mirrors real-world

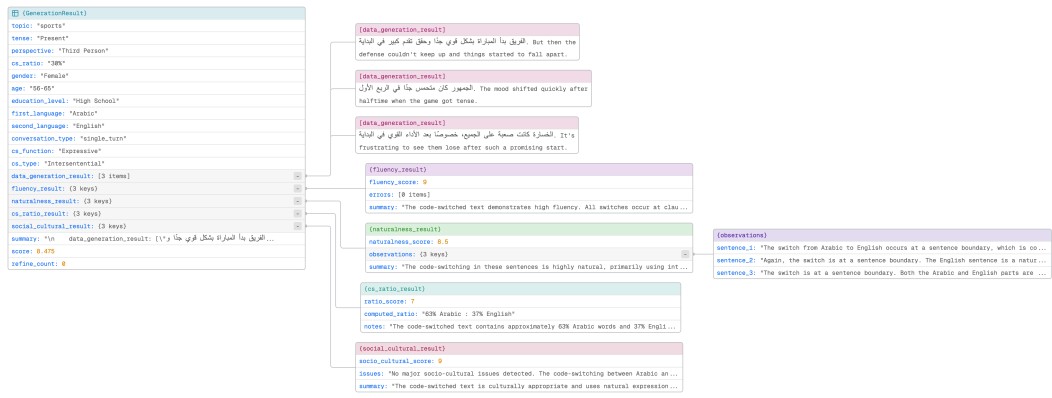

Figure A.2: Arabic-English Data Sample (with detailed meta data and evaluation results).

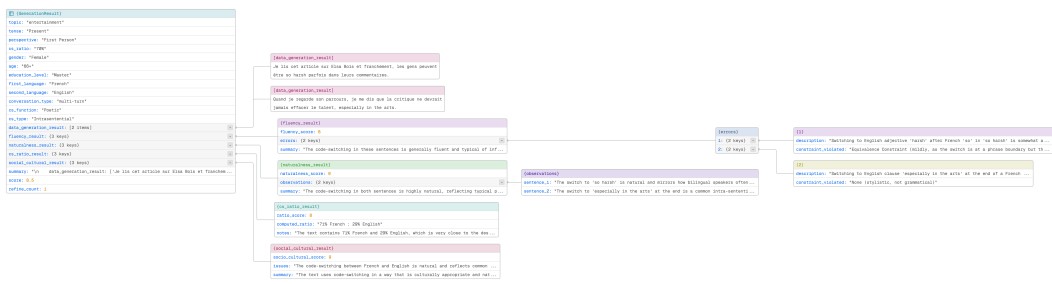

Figure A.3: French-English Data Sample (with detailed meta data and evaluation results).

discourse; (ii) rare named entities or culturally specific concepts are borrowed verbatim, boosting sociolinguistic authenticity. If the external fetch fails (network timeout or no hit), the agent falls back to the base prompt, so the graph remains side-effect free.

Any web API, local knowledge base, or custom crawler can be registered at run-time. Formally, let $\mathcal{T} = \{T_1, \ldots, T_k\}$ be the current tool set; inference simply loops over $\mathcal{T}$ and concatenates all returned snippets to the prompt. Because $\mathcal{T}$ is defined in a YAML config file, users may append *arbi-*

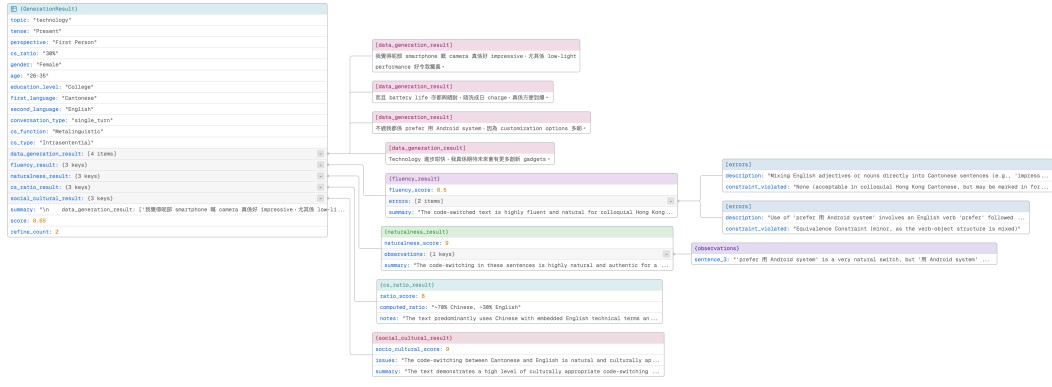

Figure A.4: Cantonese-English Data Sample (with detailed meta data and evaluation results).

| Name | Role | Type |
|------|------|------|
| GenerationAgent | Produces a CS sentence given topic, persona, matrix/embedded languages. | Generator |
| FluencyAgent | Verifies grammaticality and absence of broken morphemes. | Scorer |
| NaturalnessAgent | Estimates pragmatic plausibility with a domain-conditioned LM. | Scorer |
| CSRatioAgent | Checks whether the token-level language ratio lies within the user target. | Scorer |
| SocialCultureAgent | Validates register, borrowed lexicon, and cultural appropriateness. | Scorer |
| SummarizeAgent | Normalises individual scores and computes a weighted mean $S_{\text{final}}$. | Reducer |
| RefinerAgent | Receives failure explanations; rewrites or re-prompts the generator. | Editor |
| AcceptanceAgent | Stores accepted samples and logs metadata. | Sink |

Table B.1: Overview of the agents in the LinguaMaster, detailing their roles and types in the context of code-switching data synthesis and refinement.

*trarily many* tools without touching the LinguaMaster code base—rendering the context-enrichment stage effectively unbounded.[3]

**State-graph Execution.** Let $\mathcal{A} = \{\text{Flu}, \text{Nat}, \text{Ratio}, \text{Socio}\}$ be the evaluator set and $\tau$ the acceptance threshold. Execution proceeds as Algorithm 1. The graph is compiled once and then reused for every topic; each agent call is an LLM invocation with a task-specific prompt.

---

**Algorithm 1** LinguaMaster pipeline for one topic.

---
1: $x \leftarrow$ GENERATIONAGENT(topic, persona, params)
2: **for all** $a \in \mathcal{A}$ **do**                          ▷ parallel execution
3:     $s_a \leftarrow a(\text{x})$
4: **end for**
5: $S_{\text{final}} \leftarrow$ SUMMARIZERESULT($\{s_a\}$)
6: **if** $S_{\text{final}} \geq \tau$ **then**
7:     ACCEPTANCEAGENT(x, $\{s_a\}$)
8: **else**
9:     $x' \leftarrow$ REFINERAGENT(x, feedback=$\{s_a\}$)
10:    **goto** line 2                          ▷ one refinement loop; repeats until accepted
11: **end if**

---

**Conditional edge semantics.** In practice we encode a function `meet_criteria` predicate that materialises as a *conditional edge* in the graph compiler.[4] The predicate simply checks $S_{\text{final}} \geq \tau$. If the condition is false, control flows to `RefinerAgent`; otherwise it flows to `AcceptanceAgent`. We observed empirically that at most $1.3 \pm 0.4$ refinement iterations are required per sample.

**Scalability.** Because all evaluators are embarrassingly parallel, the graph's critical path is dominated by a single generator invocation plus aggregation— $O(1)$ per sample.

## B.2 Intuitive Understanding of Ablation Results

Instead of providing a proof sketch, we present the entire workflow along with data samples generated by each module. We believe this approach offers readers sufficient intuition to comprehend the main

---

[3]In practice we cap the payload to $\leq 4\,\text{k}$ tokens to respect the LLM context window; tools beyond that limit are sampled proportionally to their historical usefulness.

[4]Implemented via LangChain 2.0 `StateGraph`.

| Dimension | Definition | Evaluation Criteria |
|---|---|---|
| Linguistic Richness (20) | Evaluates the diversity of speech data at the linguistic level, including vocabulary, syntactic structures, and the distribution of language pairs. | Does it demonstrate a rich vocabulary and syntactic structures? Does it cover a diverse range of language pair combinations? Does it include various switching patterns (e.g., word-level switching, phrase-level switching)? |
| Language and Racial (20) Diversity | Assesses whether the dataset encompasses a combination of multiple languages, particularly representative language pairs, and whether the distribution is reasonable. | Does it include a variety of language pairs (e.g., English-Cantonese, Chinese-English, Hindi-English, etc.)? Is the distribution balanced across different language pairs, avoiding over-concentration on a few languages? Do the languages represented in the data reflect real-world usage? |
| Realism (20) | Evaluates whether the generated code-switching data aligns with real-world code-switching behaviors and whether it resembles language naturally produced by humans. | Do the sentences sound natural and conform to linguistic norms? Does it avoid obvious translationese or traces of machine generation? Is it consistent with the actual language usage patterns of the target language community? |
| Switching Naturalness (10) | Evaluates whether language switching in speech is natural and aligns with authentic code-switching behaviors. | Does the switching occur at reasonable points (e.g., at grammatically permissible switch points)? Is the flow of the switching natural, making it sound like authentic human language behavior? Does the switching serve pragmatic functions (e.g., emphasis, topic change, etc.)? |
| Contextual Coherence (10) | Assesses whether the language switching in speech is contextually appropriate and coherent. | Is the semantic flow of the content after switching smooth? Is the switching consistent with the theme of the conversation or sentence? Does it avoid semantic conflicts or logical incoherence? |
| Grammatical Accuracy (10) | Evaluates whether the language switching in speech adheres to the grammatical rules and semantic logic of both languages. | Does the switched language conform to grammatical rules? Is the semantic content clear and unambiguous? Does it avoid switches that do not comply with language norms? |
| Audio Quality (10) | Assesses whether the audio quality of the speech data is clear and free from noise, making it suitable for speech processing tasks. | Is there background noise or audio distortion present? Is the speech signal clear and easy to understand? Is it suitable for speech recognition or related tasks? |

Table B.2: Definitions and Evaluation Criteria for Dimensions of Code-Switching Corpus Quality Assessment.

results of the paper. Table B.3 provide a clearer and more intuitive understanding of the effectiveness of each module, we analyze the data samples generated by each module. Through this analysis, we aim to demonstrate the individual contributions and effectiveness of each module within the workflow, offering a comprehensive view of the data synthesis process and its results.

## B.3 Financial and Computational Cost

As shown in Table B.4, we provide a detailed breakdown of each agent's cost and their respective proportions in the workflow. With pricing based on the following structure: $5.00 per 1M input tokens, $2.50 per 1M cached input tokens, and $20.00 per 1M output tokens. On average, the cost of

| Dimension | Naive 4o | Agentic | Tool-Aumented |
|---|---|---|---|
| Original Sentence | 我好鍾意睇電影, because it's so exciting! | 我覺得呢部movie plot twist 好unexpected, 真係令我驚喜。 | 「最近有好多新劇上畫，你有無聽講過《國色芳華》？」
「有啊！不過更新得好慢，真係會好心急。」
「I know, right? 我真係好想binge-watch 一下。」 |
| Switch Type | Inter-sentential Code-Switching | Intra-sentential Code-Switching | Extra + Intra-sentential Code-Switching |
| Switch Point Legitimacy | End of sentence grammatically permitted | After noun phrase natural grammatical boundary | "I know, right?" as standalone sentence, "binge-watch" at end of Chinese sentence both legitimate switch points |
| Functional Intent | States preference | Evaluative function | Agreement marker expression of future intent |
| Syntactic/ Structural Richness | Very short main clause + single English reason clause | Moderate main clause + noun phrase + evaluative clause | Relatively long three turns including question, complaint about slow updates, agreement, and plan—highest structural richness |
| Naturalness/ Realism | Highly colloquial Hong Kong style template-like | Quite colloquial with "plot twist" phrase | Natural chat flow containing question, reaction, agreement, and personal plan reflects authentic conversation |
| Diversity of Switches | One switch fixed vocabulary | Two switches one common noun one specialized phrase | Three switches an exclamatory sentence a borrowed term a full English expression highest variety of switching |
| Possible Improvements | Extend sentence add Intra-sentential switching | Add contextual linking | Already rich add professional review phrases or more expressive language |

Table B.3: Comparison of three code-switching examples across dimensions

generating each text entry in the dataset is $0.10. In contrast, the audio data in the dataset involves hiring diverse speakers for recording, with an average cost of $0.26 per audio entry. All ASR model testing experiments were conducted on an RTX A6000 GPU to ensure optimal performance and consistency. For manual scoring and recording tasks, volunteers were compensated at an hourly rate of 450 HKD/h to ensure fairness and quality contributions to the study.

# C Assessment Details

## C.1 Human and LLM-based Evaluation Scoring Criteria

To evaluate the quality of the generated data in comparison to existing code-switching corpora, we analyze it across eight dimensions: Linguistic Richness, Language and Racial Diversity, Realism, Switching Naturalness, Contextual Coherence, Grammatical Accuracy, and Audio Quality. The first three dimensions are the most critical, each assigned a weight of 20 points, while the remaining dimensions are weighted at 10 points each, making the total score 100 points. However, as LLMs cannot evaluate Audio Quality, the maximum score for LLM-based evaluations is 90 points. Table B.2 provides detailed definitions and evaluation criteria for each dimension.

| Agent | Prompt Token | Completion Token | Prompt $ | Completion $ | Cost Proportions % |
|---|---|---|---|---|---|
| GenerationAgent | 600 | 200 | 0.00150 | 0.00200 | 35 |
| FluencyAgent | 200 | 150 | 0.00050 | 0.00150 | 14 |
| NaturalnessAgent | 220 | 200 | 0.00055 | 0.00200 | 17 |
| CSRatioAgent | 180 | 120 | 0.00045 | 0.00120 | 11 |
| SocialCultureAgent | 160 | 200 | 0.00160 | 0.00200 | 13 |
| RefinerAgent | 450 | 70 | 0.00113 | 0.00070 | 10 |
| Per item total | 1840 | 900 | 0.00461 | 0.00899 | 100 |

Table B.4: Cost and proportions of each agent in the workflow.

## C.2 Test Data Cleaning Procedures

The ASR outputs of the SenseVoice-Small and Qwen2-Audio-7B-Instruct models often contain extraneous information. For instance, SenseVoice-Small may include emojis, which can be easily removed using simple code. However, the extraneous outputs of Qwen2-Audio-7B-Instruct are more diverse and complex, requiring manual cleaning. Taking German ASR tests as an example, such extraneous outputs may include phrases like: *"Der Inhalt dieser Aufnahme lautet"*, *"Diese Person sagte"*, *"Das Audio sagt"*, *"Er sagte"*, *"The speech is in German, saying"*, *"The original content of this audio is"*, *"The person said in German"*.

# D Linguistic Analysis

In the appendix, we provide a detailed supplement to the Section 3.1.2 from the main text.

Beyond syntax, code switching is influenced by semantic content and discourse context. Bilinguals often switch languages to more precisely express a concept or nuance. A key observation is that certain ideas or terms are better conveyed in one language than the other, leading speakers to switch in order to preserve the intended meaning. For instance, speakers might use the language in which a technical term or culturally specific concept exists rather than attempt a clumsy translation. In one survey, researchers found that interviewees would speak in the formal language of schooling for topics where "certain concepts only exist in that language," but switch to their native community language for everyday matters. This illustrates how semantic domains can drive language choice. Similarly, a bilingual might start a sentence in one language and switch to quote a saying or idiom from the other language because the idiom carries a rich meaning that would be lost if translated. Here the semantic content is dictating the switch: the alternate language provides a better semantic or pragmatic fit for that segment of speech.

Myers-Scotton notes that the primary reason code switching happens at all is often because "a switch to another language better conveys the speaker's semantic and pragmatic intentions" at that moment. In a conversation, a bilingual might suddenly insert a word or phrase from L2 to capture an emotion or concept that L1 lacks an equivalent for. For example, an Arabic–English bilingual talking in English might switch to Arabic to use a word like "yalla" (which conveys encouragement/urgency roughly meaning "come on") because it succinctly expresses a nuance that English might take a sentence to explain. Such switches contribute to the expressive precision of the discourse. Semantically, code switching can also serve to clarify or emphasize meaning. If a listener doesn't understand a term in one language, the speaker may repeat or explain it in the other language (a form of elaborative code switching for clarification). Conversely, switching can mark a quotation or a punchline, signaling "this part is in another code for a reason." These choices show bilinguals leveraging the full semantic resources of their language repertoire. It's important to note that while the surface syntactic structure of a code-switched sentence follows the rules as discussed, the meaning of a code-switch often comes from the interplay of both languages. Sometimes switching itself carries meaning – for example, shifting to another language can add a different connotation or subtext. In discourse, a switch might indicate a change in topic or a shift in the speaker's stance. Gumperz (1982) famously described

conversational code switching as a discourse strategy that can function like a metaphorical "cue" to listeners – for instance, using one language for serious talk and another for humor or intimacy. In any case, the semantic effect of code switching is that it embeds the cultural and conceptual associations of the inserted language. A single word from L2 dropped into a L1 sentence can evoke the entire cultural context of L2. Thus, code switching allows bilingual speakers not just to follow grammar, but to enrich meaning, weaving both languages' semantic nuances into the conversation.

# E   Information–Theoretic Justification

We fix a probability space $(\Omega, \mathcal{F}, \mathbb{P})$. For each context $X = x$, let $P_0(\cdot \mid x)$ be the *baseline* GPT-4o distribution on utterances $Y \in \mathcal{Y}$. Successively *filter* it via constraint sets $C_1, L$ (syntax), $C_2, A = a$ (agentic), $C_3, K = k$ (facts):

$$P_{i+1}(y \mid x) = \frac{P_i(y \mid x)\mathbf{1}_{C_{i+1}}(y)}{Z_{i+1}(x)}, \qquad Z_{i+1}(x) = \sum_z P_i(z \mid x)\mathbf{1}_{C_{i+1}}(z).$$

Assume all normalisers $Z_{i+1}(x) > 0$. Let $Q(\cdot \mid x)$ be the (unknown) human distribution, with $Q(C_j \mid x) = 1$.

**Goal.**

$$H_0 > H_1 > H_2 > H_3, \tag{4}$$

$$D_{\mathrm{KL}}\big(P_{i+1}\|Q\big) < D_{\mathrm{KL}}\big(P_i\|Q\big), \quad i = 0, 1, 2. \tag{5}$$

**Lemma 1** (Conditioning lowers entropy). *If $C$ is a non-trivial event, $H(Y \mid C) < H(Y)$.*

**Lemma 2** (KL projection [46], Thm. 1). *For any $P$ and $Q$ with $\mathrm{supp}(Q) \subseteq \mathcal{C} \subseteq \mathrm{supp}(P)$, projecting $P$ onto $\mathcal{C}$ strictly decreases KL: $D_{\mathrm{KL}}(P^{\mathcal{C}}\|Q) < D_{\mathrm{KL}}(P\|Q)$ if $P(\mathcal{C}) < 1$.*

**Theorem 1** (Monotone improvement). *Under the assumptions above and whenever $P_i(C_{i+1} \mid x) < 1$ for some $x$, (4)–(5) hold.*

*Proof. Entropy.* Stage $i \rightarrow i+1$ conditions on $C_{i+1}$; Lemma 1 gives strict drop point-wise in $x$, hence for the expectation.

*KL.* Because $Q$ places zero mass outside each $C_{i+1}$, Lemma 2 applies iteratively, proving (5). □

**Practical reading.**   Lower entropy means less uncertainty / hallucination; lower KL means closer match to real code-switching data. Thus modules $\langle L, A, K \rangle$ act as *quality-improving filters*.

# F   Crowdsourcing and Participant Instructions

Participants in this study were recruited to assist with recording tasks under the following guidelines. There were no restrictions on the format of the recordings, with mp3, wav, m4a, and other formats being acceptable. Similarly, there were no limitations on the recording devices used; participants could use phones, computers, or any other suitable device. Participants were instructed to rename each recording file according to a specific naming convention: the first number represented the order of the overall context, while the second number represented the order within a multi-round conversation (e.g., "0_1" for the first turn of a conversation and "0_2" for the second turn).

Compensation for participation was offered at an hourly rate of 65 HKD, with the recorded hours including both the participants' active recording time and their resting time.

# G   Application Scenario

Our code-switching datasets, encompassing both text and audio modalities, play a pivotal role in advancing natural language and speech processing technologies in multilingual settings. In the realm of text, such datasets enable the development of machine translation systems that can seamlessly translate mixed-language content into a single target language, a necessity in social media and informal

communication. They also facilitate sentiment analysis by capturing users' emotional expressions in code-switched text, such as "The service was excelente, but it was a bit slow," where the interplay between languages presents unique challenges for traditional sentiment models. Furthermore, code-switching datasets are crucial for training chatbots and virtual assistants that can handle multilingual and contextually rich user inputs, as well as for optimizing information retrieval systems to process hybrid-language queries like "Best restaurantes cerca de mí." On the audio front, these datasets empower ASR systems to transcribe multilingual speech accurately, enabling applications such as multilingual customer service. They also support the creation of TTS systems capable of generating natural speech with smooth language transitions. Additionally, speech-to-speech translation (S2ST) systems benefit from code-switching datasets by handling scenarios where speakers alternate between languages within a single utterance. By capturing the intricacies of language mixing, code-switching datasets serve as a cornerstone for designing robust, real-world applications tailored to multilingual users.

