# OpenReview forum: "SwitchLingua: The First Large-Scale Multilingual and Multi-Ethnic Code-Switching Dataset"
_NeurIPS.cc/2025/Datasets_and_Benchmarks_Track — NeurIPS 2025 Datasets and Benchmarks Track poster_

### Official Review · Reviewer_wAm6 · 2025-07-02

**Rating:** 5
**Confidence:** 4

**Summary:**

The paper introduces LinguaMaster, a multi-agent collaboration framework to synthesize code-switching speech data. Using LinguaMaster, they construct and release SwitchLingua, a code-switching speech dataset encompassing 12 languages. Additionally, they propose SAER (semantic-aware error rate) as a novel metric for ASR evaluation.

**Additional Feedback:**

* While the paper mentions the growing demand for CSASR, CSTTS, and CLIR, it primarily focuses on CSASR. I suggest broadening the scope by adding support for CSTTS, CSIR, and language modeling (LM perplexity under code-switching).

**Dataset Code Accessibility:**

Yes

**Ethical Considerations:**

No, there are no or only very minor ethics concerns

**Final Justification:**

I am satisfied with the authors' rebuttal addressing my concern. I maintain my positive view regarding the acceptance of this paper.

**Limitations Weaknesses:**

* Lack of in-depth analysis on LinguaMaster performance
  * While the paper incorporates 12 languages, 63 ethnic groups, and 27 topics to construct SwitchLingua under LinguaMaster, it provides only overall evaluation results (Table 2). It would be interesting to identify performance differences of LinguaMaster in terms of language, ethnic groups, and topics. Will LinguaMaster be proficient in generating CS speech whose language pairs have not been sufficiently covered by publicly available data?

**Strengths Contributions:**

* Well-written paper. This paper provides a thorough review of previous studies and theories on code-switching.
* LinguaMaster is a simple yet effective framework for generating synthetic code-switching speech data.
* SwitchLingua is the first large-scale code-switching dataset.

---

> ### Author Rebuttal · Authors · 2025-07-28
>
> Thank you for recognizing the importance of our dataset. We appreciate your interest and effort in reviewing it.
>
>
> **1.More analysis experiments**
>
> Below is the experimental analysis identifying the performance differences of LinguaMaster in terms of language, ethnic groups, and topics.
>
> We selected 100 pieces of data from different languages and provided them to GPT-4o and human scorers for overall scoring, with a full score of 10.
> | Language   | LLM  Score (10 ↑)  | Human Score (10 ↑)  |
> |-|:-:|:-:|
> | Arabic-English     | 9.3       | 9.3         |
> | Cantonese-English   | 9.7       | 9.7         |
> | French-English      | 8.0         | 8.3         |
> | German-English      | 9.3       | 9.7         |
> | Hindi-English       | 9.3       | 9.3         |
> | Italian-English     | 8.0         | 8.3         |
> | Japanese-English    | 8.7       | 8.0           |
> | Korean-English      | 8.7       | 8.3         |
> | Mandarin-English    | 9.0     | 9.7         |
> | Russian-English     | 8 .0        | 8.3         |
> | Spanish-English     | 9 .3        | 9.7         |
>
> From the experiments, we observed that regions with a higher degree of code-switching tend to yield better generation quality. This indicates that the model's inherent knowledge is stronger for these regions. For instance, the Cantonese-English mixed data achieves high quality due to the prevalence of code-switching in Hong Kong.
>
> HUMAN SCORE (20 ↑) / LLM SCORE (20 ↑)
>
> |LinguaMatser Component | Linguistic Richness | Realism |
> |-|:-:|:-:|
> | No specific ethnic group | 7.3/7.0  | 7.0/ 7.3 |
> | Ethnic groups  | 9.7/8.3 | 9.0/ 7.7  |
> | No specific topics   | 8.3/7.3                            | 6.3 /7.3          |
> | Topics                 | 13.7/14.3                           | 15.7/11.7          |
>
> By defining specific topics and ethnic groups, the LinguaMaster framework can significantly enhance the richness and realism of the generated data by leveraging corresponding API tools. For example, when the topic is set to "news," integrating real-time news APIs greatly improves content authenticity.
>
> **2.LinguaMaster's ability to generate code-switching data for language pairs with limited publicly available data.**
>
> LinguaMaster demonstrates proficiency in generating code-switching data for language pairs not sufficiently covered by publicly available data.
>
> For instance, language pairs like French-English, Italian-English, and Russian-English are largely absent in public code-switching datasets. As shown in Table 2 of the paper, LinguaMaster leverages linguistic principles to guide data generation, resulting in superior performance in terms of realism and switching naturalness, even for underrepresented language pairs.
>
>
> **3.More downstream task on SwitchLingua**
>
> We extended our evaluation of SwitchLingua beyond ASR to include the TTS domain using the XTTS-v2 model [1].
>
> Specifically, we synthesized Cantonese-English code-switching audio using XTTS-v2 and compared it against human-recorded audio from our dataset. The evaluation focused on two key metrics:
>
> - Coherence: Measures the fluidity and naturalness of the synthesized audio, particularly in how well the transitions between languages are handled.
> - Accuracy: Assesses the correctness of the pronunciation, language switching, and adherence to the original text prompts. The audio samples were rated on a scale of 1 to 10 by bilingual evaluators, with the results summarized below:
> | Model | Coherence | Accuracy |
> |-|:-:|:-:|
> | XTTS-v2 | 5.0 | 4.3 |
> | Human Record | 9.3 | 9.3 |
>
> Currently, there are no strong baselines for TTS and CLIR in code-switching scenarios, largely due to the lack of high-quality datasets like ours. We hope that our dataset, SwitchLingua, can drive progress in these fields and inspire further exploration and development, and we plan to further explore the areas you highlighted in the future.
>
> [1] Casanova, Edresson, et al. "Xtts: a massively multilingual zero-shot text-to-speech model." arXiv preprint arXiv:2406.04904 (2024).

---

> > ### Comment · Reviewer_wAm6 · 2025-08-04
> >
> > Thank you for your response. I believe my initial score represents my positive view toward this paper. I will keep my score.

---

> > > ### Author Response · Authors · 2025-08-06
> > >
> > > Thank you for your time and effort in reviewing our paper.

---

### Official Review · Reviewer_5chx · 2025-07-02

**Ethics Flags:** Safety and security, Discrimination, …
**Rating:** 4
**Confidence:** 4

**Summary:**

This paper introduces SwitchLingua which is the first large-scale multilingual and multi-ethnic code-switching (CS) datasets. The SwitchLingua contains 420K CS textual samples across 12 languages, and over 80 hours of audio recordings from 174 speakers representing 18 countries/regions and 63 racial/ethnic backgrounds, based on the textual data. Furthermore, a multi-agent collaboration framework namely LinguaMaster is proposed to efficiently synthesize the code-switching datasets. The authors also propose Semantic-Aware Error Rate, incorporating semantic information for a more accurate and context-aware assessment of system performance.

**Additional Feedback:**

- Dataset not available without a consent of the authors. Need to change so that the reviewer can see those without requesting an access since the review process is a single-blinded.

**Dataset Code Accessibility:**

Yes

**Dataset Code Comments:**

The authors have provided Github and Huggingface links for the proposed dataset. They have thoroughly reported the details of the dataset in their Github page.

**Ethical Comments:**

- Safety and Security: Even though the authors point out that there is a potential risk of being misused for voice synthesis in fraudulent schemes, they have not explicitly described how will the enforcement mechanism should be, only mentioning that they would require users to sign licensing agreements or terms of use that explicitly restrict misuse.

- Bias: Mentioned in the weakness section, the textual data generated by the existing LLM is mostly English-centric. My concern is whether the LLM has correctly delivered the minority-language code-switch patterns.

**Ethical Considerations:**

Yes, there are ethics concerns that require attention by the authors

**Final Justification:**

The rebuttal resolves my concern in some point of view, so I would raise my score.

**Limitations Weaknesses:**

- The textual data generated by the existing LLM is mostly English-centric. My concern is whether the LLM has correctly delivered the minority-language code-switch patterns.

- The author proposed the dataset SwitchLingua which consists of 80 hours audio data. The 80 hours audio data seem to be modest for a 12-language benchmark, which is 6-7 hours per pair. Well-known model like Whisper already has trained with more than tens of thousands of hours, so the impact may worth on evaluation or post-training not pre-training?

- The authors only focused on the ASR performances, not exploring TTS or CLIR performance from SwitchLingua. Demonstrating at least one (or two) more downstream task would strengthen the evaluation.

**Strengths Contributions:**

- The authors have generated the first large-scale comprehensive multilingual and multicultural code-switching dataset, SwitchLingua. The proposed SwitchLingua covers over 10 languages and explicitly sample diverse ethnicities.

- The proposed Semantic-Aware Error Rate (SAER) explicitly mixes from error with the multilingual semantic similarity model LaBSE-based semantic distance and addresses well-known WER weakness in code-switching.

- The detailed experimental results show the effectiveness of the proposed LinguaMaster framework and the Semantic-Aware Error Rate (SAER). Especially, 6 competitive ASR baselines are evaluated over 12 language pairs which is extensive.

---

> ### Author Rebuttal · Authors · 2025-07-28
>
> Thank you for recognizing the importance of our dataset. We appreciate your interest and effort in reviewing it. To make the dataset more accessible, we have updated our Hugging Face repository to include some sample data for demonstration purposes. Regarding full dataset access, we will carefully review each request to ensure there are no potential security risks and will approve applications accordingly. Thank you for your patience and understanding!
>
> **1. Model bias**
>
> Although the textual data generated by existing LLMs is predominantly English-centric, it effectively captures and delivers minority-language code-switching patterns.
>
> This capability is further enhanced by our LinguaMaster framework, which is specifically designed to address these concerns by incorporating advanced linguistic principles. The LinguaMaster framework ensures contextual coherence and natural code-switching through the integration of two specialized agents:
>
> - Naturalness Agent: This agent evaluates the generated data to ensure the code-switching patterns exhibit naturalness and authenticity, aligning with how bilingual or multilingual speakers typically alternate between languages.
> - CSRatio Agent: This agent monitors the ratio of code-switching to ensure a balanced output, avoiding extremes such as insufficient or excessive switching, which could compromise the realism of the generated data.
>
> Additionally, the framework employs the MCP module to leverage external tools in real time for topic-specific information retrieval. For example, if the topic is news, the framework calls relevant news APIs to gather recent updates and ensure the generated data reflects current societal and cultural contexts.
>
> As demonstrated in Table 2 of our paper, the textual data generated by LinguaMaster excels in several key metrics: linguistic richness, language diversity, racial inclusivity, and overall realism. These factors ensure that the LLM accurately captures and reproduces minority-language code-switching patterns in a way that reflects real-world usage.
>
> To further address your concerns, consider the following German-French instance, which highlights the framework's performance:
>
> HUMAN SCORE (↑) / LLM SCORE (↑)
> |Language|Linguistic Richness|Realism|Switching Naturalness|Contextual Coherence|Grammatical Accuracy|
> |:-:|:-:|:-:|:-:|:-:|:-:|
> |German-French|12.9/13.7|15.6/17.3|6.7/7.0|8.5/7.3|9.3/9.0|
>
> From the above results, you can see how LinguaMaster effectively addresses concerns about minority-language code-switching patterns. Its robust framework ensures high-quality, linguistically rich, and contextually accurate outputs, alleviating doubts about the LLM's capabilities in this area.
>
> **2.Dataset size**
>
> The impact of the SwitchLingua dataset is significant not only for evaluation but also for pre-training, despite its modest size of 80 hours. In addition to its audio modality, our dataset also includes a text modality, which comprises 440k text samples. While the audio dataset is currently limited to 80 hours due to the high cost of recording, it already stands as the largest dataset in the code-switching domain.
>
> It is important to note that Whisper's pre-training primarily relies on single-language datasets sourced from vast, low-cost internet data. While these datasets are large, they are not optimized for code-switching scenarios and often suffer from lower quality. In contrast, the SwitchLingua dataset is specifically designed for code-switching tasks, with audio recordings created by hiring bilingual speakers with verified proficiency. This ensures both higher data quality and greater production costs, making it more specialized and impactful for code-switching benchmarks.
> As noted in Appendix A.1, the largest existing code-switching dataset prior to ours, MIAMI, consists of only 40.8 hours of audio data, and it is limited to a single code-switching scenario: Welsh-Spanish. Furthermore, MIAMI's low code-switching ratio makes it less suitable for robust evaluation or pre-training of models focused on multilingual or code-switching tasks.
> In contrast, the SwitchLingua dataset provides unprecedented diversity and scale within the code-switching domain. It spans 12 languages, offering a more comprehensive and diverse benchmark for both evaluation and pre-training. While the dataset may seem small compared to Whisper's pretraining corpus, its specialization in code-switching, combined with its high-quality design, makes it invaluable for advancing code-switching models.
>
> **3.More downstream task on SwitchLingua**
>
> We extended our evaluation of SwitchLingua beyond ASR to include the TTS domain using the XTTS-v2 model [1].
>
> Specifically, we synthesized Cantonese-English code-switching audio using XTTS-v2 and compared it against human-recorded audio from our dataset.
> The evaluation focused on two key metrics:
> - Coherence: Measures the fluidity and naturalness of the synthesized audio, particularly in how well the transitions between languages are handled.
> - Accuracy: Assesses the correctness of the pronunciation, language switching, and adherence to the original text prompts.
> The audio samples were rated on a scale of 1 to 10 by bilingual evaluators, with the results summarized below:
> | Model | Coherence | Accuracy |
> |:-:|:-:|:-:|
> | XTTS-v2 | 5.0 | 4.3 |
> | Human Record | 9.3 | 9.3 |
>
> Currently, the lack of strong baselines in TTS and CLIR for code-switching scenarios underscores the need for datasets like ours to fill this gap and enable advancements in these areas.
>
> We hope that SwitchLingua can drive progress in underexplored domains like TTS and CLIR. By providing a high-quality, multilingual, and code-switching dataset, we aim to inspire further exploration and development of robust baselines for these tasks. This would not only strengthen the evaluation of code-switching systems but also expand the practical applications of such technologies in real-world multilingual settings.
>
> **4.Safety and Security**
>
> To accommodate the single-blind review process, reviewers can directly access the sample of our dataset provided on HuggingFace, without requiring prior approval.
>
> We have made the SwitchLingua dataset available on HuggingFace under a license that ensures protection against misuse. A license template is provided, and access no longer requires direct author consent for the review process. Reviewers can now access the dataset by agreeing to the licensing terms without delays.
>
> ### **Licensing Terms**
> The licensing framework includes robust mechanisms to prevent misuse, as detailed below:
> 1. Permitted Use
> The SwitchLingua dataset is provided solely for research and non-commercial purposes. Permissible uses include:
> - Developing, evaluating, or improving Automatic Speech Recognition (ASR), Text-to-Speech (TTS), and Cross-Language Information Retrieval (CLIR) systems.
> - Academic research and publication, provided proper credit is given to the Licensor.
>
> 2. Prohibited Use
> To ensure ethical use, the Licensee agrees NOT to use the dataset for the following purposes:
> - Fraudulent or malicious activities, including but not limited to voice synthesis for impersonation, scams, or identity theft.
> - Development of deceptive systems designed to mislead individuals or organizations.
> - Commercial applications without prior written approval from the Licensor.
> - Redistribution, sublicensing, or resale of the dataset, in whole or in part.
> 3. Safety and Security Measures
> To further mitigate risks of misuse, the Licensee agrees to:
> - Use the dataset only within secure and controlled environments.
> - Implement technical safeguards to prevent unauthorized access or sharing of the dataset.
> - Disclose the intended use case to the Licensor and provide assurances of compliance with these terms.
> 4. Enforcement and Compliance
> The Licensee must comply with the following enforcement mechanisms:
> - Sign the binding Agreement that outlines all licensing terms.
> - Provide their institutional or organizational affiliation and contact details for accountability purposes.
> - Allow the Licensor to conduct periodic audits to ensure compliance with the terms.
> In the event of a breach of this Agreement, the Licensor reserves the right to:
> - Revoke access to the dataset immediately.
> - Pursue legal action, if necessary.
> - Report misuse to relevant authorities for further investigation.
>
> [1] Casanova, Edresson, et al. "Xtts: a massively multilingual zero-shot text-to-speech model." arXiv preprint arXiv:2406.04904 (2024).

---

> ### Author Response · Authors · 2025-08-06
>
> Dear Reviewer,
>
> I hope this message finds you well. As the discussion period is nearing its end. I wanted to ensure we have addressed all your concerns satisfactorily. If there are any additional points or feedback you'd like us to consider, please let us know, and we're eager to address any remaining issues to improve our work.
>
> Thank you for your time and effort in reviewing our paper.

---

> > ### Comment · Reviewer_5chx · 2025-08-07
> >
> > Thank you for the detailed rebuttal based on my review. I think the rebuttal resolves my concern in some point of view, so I would raise my score.

---

> > > ### Author Response · Authors · 2025-08-08
> > >
> > > Thank you for your time and effort in reviewing our paper.

---

### Official Review · Reviewer_ukaa · 2025-07-03

**Rating:** 4
**Confidence:** 3

**Summary:**

This paper introduces SwitchLingua, a large-scale and multi-ethnic code-switching synthetic dataset, covering both textual and audio modalities. According the proposed framework (dubbed "LinguaMaster"), this is a purely synthetic dataset that is generated through collaborations of a few GPT-4o-based agents (Generation, refiner, summarize, acceptance etc.). The resulting dataset achieved higher quality than previously available datasets (as measured by both human and LLM evaluations), thanks to components like multi-agent collaboration, incorporating linguistic principle in the prompt, as well as integrating tools. The paper also showed that the SwitchLingua can serve as a multilingual code-switching ASR benchmark.

**Additional Feedback:**

* A few floating elements (Figure 2, Table 2/3) are 3 pages away from where they were first mentioned in the text.
* A lot of space was spent on the specifics of the linguistic principles (Section 3.1), which would have been better utilized with some basics of the dataset (contents in Appendix A1-A3, especially A3).
* In general, there could have been a better structure and transition across the three subsections of Section 3 -- right now they are quite disconnected from each other and it is not clear to me how certain parts (esp. SAER metric) are motivated/used.
* L233: Experimental Stepups -> Experimental Setup

## Questions

* Could you please explain what's the goal of SAER in the data creation process other than demonstrating that SwitchLingua can serve as a code-switching ASR benchmark? How was the hyper-parameter tuned?
* The multi-modality of the dataset seems to be a good feature. However, I was not able to find clear information regarding the relationship between the textual/audio part of the dataset -- was the text part generated first, then read by the speakers?
* Can you provide some information regarding the human evaluators? Were they also involved in the audio data creation process? What is the selection process for those evaluators?

**Dataset Code Accessibility:**

Partly

**Dataset Code Comments:**

I was able to access the code, but access to the data requires approval from repository owners and was still pending when I wrote this review.

**Ethical Considerations:**

No, there are no or only very minor ethics concerns

**Final Justification:**

I have read the other reviews and agree with the other reviewers that this is a technically-solid paper and the dataset is useful (as i have clearly stated in my response). Regarding the two points raised in the final author remarks, I understand the importance of dependency parsing to the proposed dataset creation approach, but that's not the point I'm having issues with. The overarching issue is that the current status of the paper seriously limits my ability as a reader to independently understand the contribution of the paper, and a lot of questions were only clear after discussion with the author.

But since the dataset is technically solid and all other reviewers don't seem to have the same issue, I'll yield and improving my score to 4.

I hope the authors make an effort to improve their paper quality as they have promised. This will eventually help your work to be seen and used by the wider research community. Thanks in advance for the effort!

**Limitations Weaknesses:**

* The presentation quality of this paper is rather poor, and could use a significant overhaul. (See additional feedback for details)
* Some of the data synthesis/evaluation setup are flawed:
  * L193 -- I'm not sure parsing $L_1$ into dependency tree is necessary
  * Using the same model (GPT-4o) for data generation and evaluation introduces self bias
  * The newly introduced SAER metric has a tunable parameter $\alpha$ for which no source could be found as to how this is set

**Strengths Contributions:**

* To the best of my knowledge, the wide coverage of the dataset (multilingual + multi-ethnic + multi-modal) is unprecedented.
* Human evaluation shows that the data quality is higher than quite a few other datasets.

---

> ### Author Rebuttal · Authors · 2025-07-28
>
> Thank you for recognizing the importance of our dataset. We appreciate your interest and effort in reviewing it. To make the dataset more accessible, we have updated our HuggingFace repository to include some sample data for demonstration purposes. Regarding full dataset access, we will carefully review each request to ensure there are no potential security risks and will approve applications accordingly. Thank you for your patience and understanding!
>
>
> **1. Minor writing mistakes**
>
> Mistakes like floating elements and logic flow in Section 3.1 will be fixed in the final version to significantly improve the quality and clarity of the paper.
>
>
> **2. Necessary for the dependency tree**
>
> Modern theories of code‑switching emphasise that language mixing is not arbitrary but constrained by grammar. Two widely cited constraints—the Free‑Morpheme constraint and the Equivalence constraint—state that a switch can occur only at a point where neither side of the boundary is a bound morpheme and where the surface word‑order of the two languages match, respectively. Further, the Functional Head Constraint (FHC) stipulates that a switch cannot occur between a functional head (e.g., complementiser, determiner, inflection) and its complement. Enforcing these constraints requires knowledge of the internal syntactic relationships within the L1 sentence—for example, to know which determiner belongs to which noun, or to identify the scope of a verb phrase. Token‑by‑token heuristics are insufficient for this because they do not distinguish between lexical material and functional heads and cannot tell whether a switch breaks a phrase.
>
> A dependency tree provides exactly the hierarchical structure needed to apply these constraints. Dependency parsing makes head–dependent relations explicit; each node records which word is the syntactic head and which words are its dependents, forming a graph that represents the sentence’s syntax. For example, use dependency parsing to extract relations such as nominal subject and direct object and determine clause boundaries; they note that dependency parsing represents “semantic relations between words in a sentence” and rely on it to “identify various segments appropriately”. By traversing the parse tree, they ensure that clauses and adjuncts are segmented in a way that preserves semantic information. Similarly, in our work, the parse allows us to locate switchable spans that respect the Free‑Morpheme and Equivalence constraints and to ensure that no switch cuts between a functional head and its complement. Without this structural information, the generator could inadvertently switch inside a determiner–noun unit or a verb–object relation, violating FHC and producing ungrammatical outputs.
>
> Without access to an explicit dependency tree the LLM tends to place the switch at whichever token boundary looks “safe” from a surface perspective, but this ignorance of underlying head–dependent structure regularly produces the following failure modes:
>
> • Arabic ↔ English: switching inside the bound-pronoun cluster “كتبت-ه” (I-wrote-it) so that the stem “كتبت” is rendered in English while the pronominal suffix “-ه” is left in Arabic.  The result violates the Free-Morpheme constraint because the bound morpheme “-ه” is separated from its host.
>
> • German ↔ English: splitting the verb phrase “hat gegessen” / “has eaten” and moving only “eaten” across the boundary.  Because German keeps the auxiliary in second position and the participle at the end, the resulting mixed clause breaks the Equivalence constraint—the head–dependent (aux-participle) order no longer matches in the two languages.
>
> • Cantonese ↔ English: detaching the English article “a” inside the noun phrase “我買咗 a 鞋” (“I bought a shoe”).  Here the functional head (the determiner) is separated from its complement noun, violating the Functional-Head constraint.
>
> Each error stems from the model’s lack of syntactic awareness and directly contravenes at least one of the canonical constraints.  By inserting the dependency-parsing step we give the generator a deterministic map of head–dependent relations, allowing it to rule out any candidate span that would leave a bound morpheme orphaned, disturb head order, or sever a functional head from its complement.  The parser is therefore an essential safeguard that prunes invalid switch points before the LLM ever produces text, keeping the search space grammatically sound and sharply reducing the number of downstream refinement cycles.
>
> **3. Evaluation model self-bias**
>
> We use GPT-4o to generate data, and Qwen2.5-32B-Instruct for evaluation to avoid self-bias.
>
> |Dataset|Linguistic Richness|Language and Racial Diversity|Realism|Switching Naturalness|Contextual Coherence|Grammatical Accuracy|Overall|
> |:-:|:-:|:-:|:-:|:-:|:-:|:-:|:-:|
> |MDCC |6.1|5.2|5.0|6.1|4.1|4.3|30.8|
> |CSLR|6.2|5.1|6.2|6.0|6.5|6.5|36.5|
> |ASCEND|5.7|5.4|7.1|5.2|6.3|6.2|35.9|
> |MCE|10.4| 6.2|11.1|7.4|7.3|7.3|49.7|
> |SEAME|12.1|7.2|10.2|7.3|7.2|7.2|51.2|
> |ArzEn|8.9|6.8|10.7|7.5|6.4|6.1|46.4|
> |BangorTalk|9.7|8.0|11.0|6.5|6.7|6.2|48.1|
> |MediaParl|9.0| 5.0|6.6|7.2|7.7|7.5|43.0|
> |SwitchLingua|15.2|19.7|14.3|8.3|9.5|9.7|76.7|
>
> From the table, it is evident that SwitchLingua stands out as the top-performing dataset, consistently surpassing others across multiple dimensions. In particular, it excels in Linguistic Richness (15.2), Language and Racial Diversity (19.7), and Realism (14.3). These scores are significantly higher compared to other datasets, underscoring SwitchLingua's strength in capturing diverse, realistic, and linguistically rich content.
>
> **4.Meaning and Function of SAER metric**
>
> The SAER metric is a novel evaluation approach specifically designed for code-switching ASR tasks. Traditional ASR metrics, such as WER or CER, are primarily developed for single-language ASR tasks and fail to capture the unique challenges of code-switching scenarios.
>
> For example, consider the Cantonese-English code-switching sentence shown in the image: “我今日有一堂课㗎final project要present, 啲slides仲未搞A掂.”
>
> When processed by a general ASR model like Whisper, which is trained on multiple languages, the output often defaults to a single language. For instance, the model might produce a purely English transcription: “I have a final project to present for one of my classes today, but the slides aren't done yet,” or a purely Cantonese transcription: “我今日有一堂课㗎，啲幻燈片仲未搞掂.”
>
> Using traditional ASR metrics like WER or CER in such cases only calculates word- or character-level errors, completely disregarding the semantic content and language switches in the text. This limitation makes it impossible to accurately evaluate the ASR performance in code-switching contexts. For instance, in Cantonese, “项目” means “project” and “展示” means “present,” but these semantic equivalents are ignored by traditional metrics.
>
> To address this, we introduced SAER, which integrates semantic similarity into ASR evaluation. Specifically, we leverage the LaBSE model to compute the semantic similarity between the reference and predicted texts. By combining this semantic evaluation with traditional ASR metrics, SAER provides a more comprehensive measure of ASR accuracy in code-switching scenarios.
>
> **5. SAER Tunable parameter α setting**
>
> We determine the final α value by ensuring that the SAER aligns closely with human evaluation scores, confirming consistency between the two metrics. When α =0.5, the model performance ranking and error rate are most realistic.
> | α | Qwen2-Audio-7B-Instruct | Qwen2-Audio-7B-Instruct (clean) | Seamless M4T Large v2 | SenseVoice-Small | Wav2Vec2-XLSR-53 | Whisper Large v3 |
> |:-:|:-:|:-:|:-:|:-:|:-:|:-:|
> |0|0.0770|0.0615|0.0654|0.0332|0.1754|0.0463|
> |0.3|0.1919|0.1671|0.1650| 0.0964|0.3127|0.1475|
> |0.5|0.2685|0.2376|0.1386| 0.2314|0.4042|0.2187|
> |0.7|0.3450|0.3080|0.2979| 0.1808|0.4957|0.2825|
> |1|0.4599|0.4137|0.3975|0.2440|0.6330|0.3838|
>
> In the Cantonese-English code-switching scenario, human scores indicate the following order of model performance: seamless m4t large v2 performs best, followed by whisper large v3, Qwen2-Audio-7B-Instruct (clean), Qwen2-Audio-7B-Instruct, and finally Wav2Vec2-XLSR-53. The error rates for these models are consistent with the human evaluation scores, confirming the ranking.
>
> **6.Dataset Modality order**
>
> As illustrated in Appendix A of paper, initially, we utilized our LinguaMaster framework to generate text data, then we engaged bilingual native speakers to record audio, ensuring the dataset's quality.
>
> **7.Information regarding the human evaluators**
>
> Human evaluators comprised three distinct cohorts:
> (i) Native speakers residing in target regions, for Cantonese and English code-switching (e.g., Hong Kong and Guangdong Cantonese speakers),
> (ii) Expatriates in Hong Kong originating from relevant countries (e.g., Germany, Japan), and
> (iii) Advanced-language students (≥3 years of study) from Peking University specializing in the target languages.
>
> Roles in Data Creation & Evaluation:
>
> Evaluators participated in a dual-phase process (Reviewed and annotated raw audio data, flagging sensitive content and recommending linguistic refinements).
>
> Selection Criteria:
>
> Cantonese: Exclusively native Hong Kong/Guangdong speakers.
> Other Languages:
>   - Peking University students minimum 3 years’ language specialization (Peking University's language department is one of the best in China for linguistics.), or
>   - Professionally active native speakers of the corresponding languages.

---

> > ### Comment · Reviewer_ukaa · 2025-08-01
> >
> > Thanks for the detailed response. The authors have made good clarifications regarding a few key points, including self-bias for evaluation, SAER metric, as well as data modality order. I think including those extra discussions will significantly improve the clarity and soundness of the paper. I'm still not entirely convinced that the inclusion of an intermediate dependency parsing step will do more good than harm (because of potential parsing errors and propagation to subsequent steps), but other than that my other concerns for the methodology is resolved.
> >
> > However, I still think the overall presentation quality of the paper requires a major overhaul to clear the Neurips acceptance threshold, so I'm not comfortable raising the score to a level that indicates inclination of acceptance. This is the main reason why I maintain my score. The area chair is free to override me if they think I mis-interpreted the presentation standard for this track.

---

> > ### Author Response · Authors · 2025-08-02
> >
> > Thanks for your reply.
> >
> > **1.Dependency Parsing as an Auxiliary Tool**
> >
> > The inclusion of the dependency parsing step is intended to provide additional syntactic structure as auxiliary information to the model. It serves as a supportive tool rather than a critical dependency in the pipeline. Although parsing errors may occur, they do not significantly impact the final results, as the primary generative capability of the large model remains intact, and no parsing errors were found in the data generation process.
> >
> > Additionally, our framework includes an evaluator agent, which not only assigns scores but also generates evaluation reports. If dependency parsing were causing potential errors that propagate to subsequent steps, these issues would be reflected in the evaluator agent's feedback. However, no such problems have been observed, further demonstrating the reliability of our approach.
> >
> > Moreover, as shown in Table 2 of the main text, the quality of the generated data is already very high, which highlights the robustness of our method, even with the dependency parsing step included.
> >
> > **2.Dataset Track Focus on Dataset Significance**
> >
> > We believe that the evaluation of submissions in the dataset track should prioritize the importance and impact of the dataset itself. As you noted, the wide coverage of our dataset—spanning multilingual, multi-ethnic, and multi-modal dimensions—is unprecedented and represents a significant contribution to the research community. We hope that writing quality, while important, does not override the value of the dataset in determining acceptance.
> >
> > We are committed to improving the presentation quality of our paper in future revisions and would greatly appreciate it if you could reconsider your assessment in light of the dataset’s importance and novelty.

---

> > ### Author Response · Authors · 2025-08-06
> >
> > Dear Reviewer,
> >
> > I hope this message finds you well. As the discussion period is coming to an end, I wanted to ensure that my response has fully addressed your concerns regarding the Dependency Parsing. From your comments, I understand that your main concern with our paper lies in some writing issues.
> >
> > We will carefully address them in the camera-ready version, strictly following your suggestions. Given the importance of the dataset that you have recognized, we hope these small issues will not overshadow its contribution.
> >
> > Thank you again for your valuable time and thoughtful feedback. I would greatly appreciate hearing your thoughts.

---

### Official Review · Reviewer_khsX · 2025-07-20

**Rating:** 5
**Confidence:** 4

**Summary:**

This paper introduces SwitchLingua, the first large-scale multilingual and multi-ethnic code-switching dataset, accompanied by a new data synthesis framework called LinguaMaster. The dataset covers 12 languages, providing both text and audio data. SwitchLingua aims to fill the gaps left by previous code-switching corpora by focusing on linguistic diversity and naturalness, and it includes a new evaluation metric, Semantic-Aware Error Rate (SAER), that better captures semantic accuracy in automatic speech recognition (ASR) for code-switching scenarios.

**Dataset Code Accessibility:**

Yes

**Ethical Considerations:**

No, there are no or only very minor ethics concerns

**Final Justification:**

Most concerns have been addressed. Therefore, I recommend to accept this work.

**Limitations Weaknesses:**

- The dataset’s audio portion, while large, is still artificially synthesized in part and may not fully capture the natural code-switching in daily life.
- Although the dataset is extensive, certain languages or ethnic groups may still be underrepresented relative to their real-world usage, limiting true global coverage.

**Strengths Contributions:**

This paper addresses longstanding bottlenecks in code translation research: data scarcity, generation quality, and insufficient evaluation methods. The innovation lies in its skillful integration of established linguistic theories into a multi-agent large language model workflow, thereby generating synthetic data that is more reasonable both in terms of grammar and sociolinguistics. Specifically, this paper contributes well in:
- SwitchLingua brings a much broader linguistic and ethnic diversity than existing code-switching datasets, both in language coverage and cultural representation.
- The paper provides extensive human and LLM-based evaluations, showing clear advantages in linguistic richness and realism over previous benchmarks.
- The new semantic-aware evaluation metric directly addresses the main shortcomings of traditional error rates for code-switching tasks.

---

> ### Author Rebuttal · Authors · 2025-07-28
>
> Thank you for recognizing the importance of our dataset. We appreciate your interest and effort in reviewing it.
>
> **1. The natural code-switching in daily life**
>
> Our LinguaMaster framework can capture the natural code-switching in daily life.
>
> The LinguaMaster framework integrates linguistic principles to ensure contextual coherence and switching naturalness in generated data. To achieve this, it incorporates two specialized agents: the Naturalness Agent and the CSRatio Agent. The Naturalness Agent evaluates whether the generated data truly exhibits switching naturalness, while the CSRatio Agent monitors the code-switching ratio to avoid either insufficient or excessive code-switching. Additionally, the framework uses the MCP module to call external tools in real-time to gather topic-specific information. For instance, when the topic is news, it calls news APIs to retrieve recent updates, ensuring the generated data aligns with current societal contexts.
>
> To evaluate the generated data, both LLMs and human raters score the outputs. The human raters are bilingual native speakers of the respective code-switching languages, ensuring evaluation accuracy. As shown in Table 2 of the paper, the generated data achieves high scores in Linguistic Richness, Realism, Switching Naturalness, and Contextual Coherence, indicating that our framework effectively captures the natural patterns of code-switching in daily life.
>
> **2. Coverage of languages and ethnic groups**
>
> SwitchLingua already covers the most prevalent and representative code-switching scenarios, making it well-suited for capturing the linguistic diversity observed in multilingual communities worldwide.
>
> For instance, in Hong Kong, Cantonese-English and Mandarin-English code-switching are highly common, reflecting the multilingual nature of the region. These patterns are particularly evident among key demographics such as Hong Kong locals, Cantonese speakers from Guangdong, and overseas Chinese communities, who seamlessly blend languages in both formal and informal settings. Similarly, at the US-Mexico border, Spanish-English code-switching is widely prevalent, influenced by the cultural and linguistic exchange that has evolved over decades of coexistence and interaction. These examples represent some of the most iconic and impactful cases of code-switching in a globalized world.
>
> In future work, we plan to further expand SwitchLingua’s capabilities by incorporating a broader range of languages and scenarios. By doing so, we aim to achieve a more comprehensive global representation, ensuring that SwitchLingua continues to support and reflect the dynamic linguistic practices of diverse communities.

---

> > ### Comment · Reviewer_khsX · 2025-08-06
> >
> > Thanks for detailed response. I will maintain the positive score.

---

> > > ### Author Response · Authors · 2025-08-06
> > >
> > > Thank you for your time and effort in reviewing our paper.

---

### Note · Authors · 2025-08-12

Dear Reviewers, ACs, and SACs,

We sincerely thank all of you for your efforts during the rebuttal and discussion phases. However, we would like to provide further clarification regarding some of the feedback from **Reviewer ukaa** to ensure any remaining concerns are fully resolved.

**The Dependency Parsing Question**

Reviewer ukaa raised an additional question regarding dependency parsing. While we have already provided a detailed response in the rebuttal, we did not receive further feedback, so we are unsure if there are still any unresolved concerns. To clarify further:

The inclusion of the dependency parsing step is intended to provide additional syntactic structure as auxiliary information to the model. It serves as a supportive tool rather than a critical dependency in the pipeline. Although parsing errors may occur, they do not significantly impact the final results, as the primary generative capability of the large model remains intact, and no parsing errors were found in the data generation process. Additionally, our framework includes an evaluator agent, which not only assigns scores but also generates evaluation reports. If dependency parsing were causing potential errors that propagate to subsequent steps, these issues would be reflected in the evaluator agent's feedback. Moreover, as shown in Table 2 of the paper, the quality of the generated data is already very high, which highlights the robustness of our method, even with the dependency parsing step included.

**The Presentation and Writing Quality**

Reviewer ukaa also raised concerns about the presentation and writing quality. We would like to explain that the detailed discussion of code-switching and its linguistic principles in the method section was included to provide necessary background knowledge for readers who might not be familiar with these concepts. Without this information, understanding our methodology and its motivations might be challenging. Furthermore, we have provided a more comprehensive explanation in the appendix to enhance the reader’s understanding of our work and the significance of the dataset. We believe this writing approach is reasonable and ensures clarity and accessibility for a broader audience.

We are open to further feedback and willing to adjust the content layout in future revision. Finally, we believe SwitchLingua, as a multilingual, multi-ethnic, and multi-modal dataset, holds the potential to advance research in the code-switching related field.

---

### Decision · Program_Chairs · 2025-09-18

**Decision:**

Accept (poster)

**Comment:**

This paper introduces SwitchLingua which is a dataset  of multilingual and multi-ethnic code-switching (CS) conversations. Code-switching is the practice of shifting between languages within the same conversation or sentence, something that multilingual people often do for various reasons. There is no prior dataset on code-switching of similar size so this is a valuable contribution.
The data has 420K text samples in 12 languages and 80 hours of audio recordings from 174 speakers representing 18 countries.
The authors then measure ASR perfomances in different languages and the effects of CS on ASR at a massive scale.

The biggest concern is that the dialogs in the dataset are not collected from natural human conversations with spontaneous code-switching. Instead, they were synthetically generated first as text using the authors’ LinguaMaster multi-agent framework (which applies linguistic, semantic, and sociolinguistic constraints), and then the audio was recorded by human speakers reading these generated dialogs aloud. This is a major limitation since the synthetic scripts may have all kinds of biases (beyond the English-centric bias that was discussed). In any case, the value for a massive multilingual data is significant so overall I think this paper deserves to be published.